# Flow-Based Single-Step Completion for Efficient and Expressive Policy Learning

**Prajwal Koirala & Cody Fleming**
Iowa State University
Ames, Iowa, USA
`{prajwal, flemingc}@iastate.edu`

## Abstract

Generative models such as diffusion and flow-matching offer expressive policies for offline reinforcement learning (RL) by capturing rich, multimodal action distributions, but their iterative sampling introduces high inference costs and training instability due to gradient propagation across sampling steps. We propose the *Single-Step Completion Policy* (SSCP), a generative policy trained with an augmented flow-matching objective to predict direct completion vectors from intermediate flow samples, enabling accurate, one-shot action generation. In an off-policy actor-critic framework, SSCP combines the expressiveness of generative models with the training and inference efficiency of unimodal policies, without requiring long backpropagation chains. Our method scales effectively to offline, offline-to-online, and online RL settings, offering substantial gains in speed and adaptability over diffusion-based baselines. We further extend SSCP to goal-conditioned RL, enabling flat policies to exploit subgoal structures without explicit hierarchical inference. SSCP achieves strong results across standard offline RL and GCRL benchmarks, positioning it as a versatile, expressive, and efficient framework for deep RL and sequential decision-making.

## 1 Introduction

Learning effective policies from fixed datasets, without active environment interaction, is a central challenge in offline reinforcement learning (RL) and related fields. In these settings, the choice of policy parametrization plays a pivotal role in determining the agent's ability to generalize and perform well in complex environments. While unimodal Gaussian policies have remained the de facto standard due to their simplicity and compatibility with gradient-based learning algorithms, their limited expressiveness often hampers performance in multimodal or highly non-linear behavior distributions (Fu et al., 2020; Lange et al., 2012; Wang et al., 2022; Tarasov et al., 2023).

Recent advances in generative modeling, particularly diffusion and flow-based methods, have introduced significantly more expressive alternatives to traditional unimodal policy representations by enabling the modeling of complex, multimodal action distributions (Ho et al., 2020; Song et al., 2020; Wang et al., 2022; Lipman et al., 2022; Liu et al., 2022b). Diffusion-based policies, in particular, have demonstrated impressive performance in imitation learning (IL) settings, especially when learning from expert human demonstrations (Chi et al., 2024; Pearce et al., 2023). Despite these advantages, practical challenges like computational inefficiency at inference time have limited the broader adoption of such generative policy classes. For instance, diffusion models typically require tens to hundreds of denoising steps to generate a single action, making them less practical for real-time control (Ding & Jin, 2023; Park et al., 2025). Moreover, there is the difficulty of integrating iterative generative structures into conventional off-policy value-based learning frameworks. Their reliance on long-generation horizons renders backpropagation through time (BPTT) either intractable or highly inefficient, impeding scalable off-policy gradient-based optimization in offline RL. Nonetheless, recent empirical studies suggest that behavior-constrained policy gradient methods (such as DDPG+BC and TD3+BC (Fujimoto & Gu, 2021)) consistently outperform weighted behavior cloning approaches like Advantage Weighted Regression (AWR), due to their ability to both

---

The code is available at `https://github.com/PrajwalKoirala/SSCP-Single-Step-Completion-Policy`.

better exploit the value function and somewhat extrapolate beyond the coverage of dataset actions (Park et al., 2024b).

While recent efforts have explored mitigation strategies, such as reducing the number of generative steps or distilling policies into simpler networks, these approaches often introduce trade-offs in empirical performance, sensitivity to hyperparameters, or increased architectural complexity (Ding & Jin, 2023; Janner et al., 2022; Park et al., 2025). Motivated by these limitations, we propose *flow-based single-step completion* as a powerful policy representation that combines the expressiveness of generative models with the efficiency and simplicity required for scalable learning across offline reinforcement learning, imitation learning, and broader robotic decision-making tasks. The mechanism is illustrated in Figure 1, where *completion vectors* provide a one-step generative shortcut to the target action distribution, bypassing the need for iterative transport. By directly predicting a completion vector toward the target action in a single generative step, our method *Single-Step Completion Policy (SSCP)* circumvents the inefficiencies of sequential generation, enables compatibility with standard value-based RL algorithms, and offers a more practical and scalable solution for high-performance policy learning.

Beyond one-shot action generation, we further explore the role of *completion modeling* in hierarchical decision-making. Hierarchical methods in long-horizon goal-conditioned RL (GCRL) exploit subgoal structures by decomposing tasks into multiple decision layers. This naturally raises the question: just as *SSCP* compresses multi-step generation into a single-step policy without sacrificing performance, can multi-level hierarchical policies also be compressed into a single flat decision step while retaining their subgoal-exploiting strengths? Motivated by this, we extend shortcut/completion modeling to GCRL, yielding *GC-SSCP*, a flat inference policy that preserves subgoal benefits and significantly outperforms flat baselines, surpassing hierarchical state-of-the-art methods such as HIQL (Park et al., 2023) on average in OGBench (Park et al., 2024a) tasks.

Our key contributions are as follows:

- **A novel policy method based on single-step completion** using flow-matching generative models. This policy class is expressive and tractable, enabling direct action sampling without iterative inference. (Section 3.2)

- **Compatibility with off-policy policy gradient methods**, such as DDPG+BC without worrying about BPTT. Our approach yields strong empirical results on standard D4RL benchmarks and facilitates seamless offline-to-online finetuning. (Section 3.3)

- **A framework for distilling hierarchical behavior into flat policies.** We show how the single-step completion principle can be extended to exploit subgoal structures in long-horizon, sparse-reward, multi-goal conditioned environments. (Section 4.2)

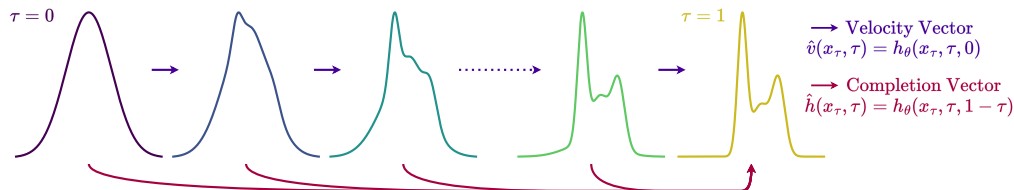

Figure 1: Depiction of completion-based flow matching: while velocity vectors propagate along the generative path, completion vectors enable shortcut one-step jumps to the target distribution. This forms the basis of our Single Step Completion Policy (SSCP) used in offline RL and related problems.

## 2 PRELIMINARIES

### 2.1 OFFLINE REINFORCEMENT LEARNING

Reinforcement learning (RL) seeks a policy $\pi$ that maximizes expected return in a Markov Decision Process (MDP) $(\mathcal{S}, \mathcal{A}, \mathcal{P}, \mathcal{R})$, where $V^\pi(s_0) = \mathbb{E}_\pi[\sum_{t=0}^{T} \gamma^t r_t \mid s_0]$. Offline RL considers the

setting where learning must occur solely from a fixed dataset $\mathcal{D} = \{(s, a, r, s')\}$ collected by an unknown behavior policy $\pi_\beta$, precluding further environment interaction.

A central challenge in this setting is the distributional shift between the learned policy $\pi$ and $\pi_\beta$, which can result in erroneous value estimates when $\pi$ selects out-of-distribution (OOD) actions (Fujimoto et al., 2019; Kumar et al., 2020; Levine et al., 2020). To mitigate this, behavior-constrained approaches regularize $\pi$ to remain close to $\pi_\beta$, often via a divergence constraint:

$$\max_\pi \; V^\pi(s), \quad \text{s.t. } D_{\mathrm{KL}}(\pi \| \pi_\beta) \leq \delta. \tag{1}$$

A widely used formulation for enforcing such constraints combines off-policy policy gradient methods (like Lillicrap et al. (2015); Fujimoto et al. (2018)) with Behavior Cloning (BC) regularization, optimizing:

$$\mathbb{E}_{(s,a)\sim\mathcal{D}} \left[ Q(s, \pi(s)) + \alpha \log \pi(a \mid s) \right], \tag{2}$$

where $\alpha$ balances exploitation and conservatism (Fujimoto & Gu, 2021; Wang et al., 2022). This serves as the foundation for our method discussed in section 3.3.

As an extension, in section 4, we also consider offline goal-conditioned RL (Andrychowicz et al., 2017; Liu et al., 2022a; Park et al., 2024a), where policies and rewards are aditionally conditioned on a goal state $g \in \mathcal{G}$. This setting often employs sparse rewards of the form $r(s, a, g) = -\mathbb{1}(s \neq g)$, encouraging the agent to reach the goal without requiring dense or hand-crafted signals.

## 2.2 DIFFUSION MODELS AND FLOW MATCHING.

**Diffusion models** learn to reverse a stochastic forward process that gradually corrupts data with noise (Sohl-Dickstein et al., 2015; Ho et al., 2020; Song et al., 2020). The forward process is typically defined as a stochastic differential equation (SDE):

$$d\mathbf{x}_\tau = f(\tau)\mathbf{x}_\tau \, d\tau + g(\tau) \, d\mathbf{w}_\tau, \tag{3}$$

where $\mathbf{w}_\tau$ is standard Brownian motion, and $f(\tau), g(\tau)$ govern the drift and diffusion schedules. The reverse-time dynamics, learned via denoising score matching, allow for approximate sampling by numerically solving a reverse-time SDE or its deterministic counterpart, the probability flow ODE. Most implementations train a neural network to predict the noise added at each timestep, minimizing a weighted mean squared error between the predicted and true noise vectors.

**Flow matching** offers an alternative that bypasses stochastic dynamics by directly modeling deterministic transport between a base distribution and the data distribution (Lipman et al., 2022; Liu et al., 2022b). Let $p_0(\mathbf{z}) = \mathcal{N}(\mathbf{0}, \mathbf{I})$ be a standard normal distribution, and $p_{\mathrm{data}}(\mathbf{x})$ the empirical data distribution. Flow matching defines a continuous interpolation path:

$$\mathbf{x}_\tau = (1 - \tau)\mathbf{z} + \tau\mathbf{x}, \quad \tau \in [0, 1], \tag{4}$$

where $\mathbf{z} \sim p_0$, $\mathbf{x} \sim p_{\mathrm{data}}$, and learns a time-dependent velocity field $\mathbf{v}_\theta(\mathbf{x}_\tau, \tau)$ that maps points along this path. The optimal velocity field under this scheme is simply the displacement vector: $\mathbf{v}^*(\mathbf{x}_\tau, \tau) = \mathbf{x} - \mathbf{z}$. The flow matching objective minimizes the expected squared deviation from this ideal field:

$$\mathcal{L}_{\mathrm{FM}} = \mathbb{E}_{\tau \sim p(\tau), \, \mathbf{z} \sim p_0, \, \mathbf{x} \sim p_{\mathrm{data}}} \left[ \| \mathbf{v}_\theta(\mathbf{x}_\tau, \tau) - (\mathbf{x} - \mathbf{z}) \|_2^2 \right], \tag{5}$$

Sampling from the learned model requires solving the ODE:

$$\frac{d\mathbf{x}_\tau}{d\tau} = \mathbf{v}_\theta(\mathbf{x}_\tau, \tau), \quad \mathbf{x}_0 \sim p_0, \tag{6}$$

using standard numerical solvers such as Euler or Runge-Kutta methods. While diffusion and flow-based models excel at capturing multimodal distributions and generating high-fidelity samples, they require several iterative steps during inference, making them impractical for real-time control applications.

## 2.3 BEHAVIOR CLONING, DIFFUSION POLICIES, AND FLOW-BASED POLICIES

**Behavior Cloning (BC)** formulates policy learning as supervised learning from expert demonstrations. Given a dataset of state-action pairs $(s, a) \sim \mathcal{D}$, the policy $\pi_\theta(a \mid s)$ is trained to maximize the likelihood of expert actions:

$$\mathcal{L}_{\mathrm{BC}} = \mathbb{E}_{(s,a)\sim\mathcal{D}} \left[ -\log \pi_\theta(a \mid s) \right]. \tag{7}$$

Recent works have explored the use of diffusion models as expressive policy classes in BC and RL (Janner et al., 2022; Pearce et al., 2023; Ding & Jin, 2023; Wang et al., 2022; Chi et al., 2024). Instead of directly predicting actions, the model learns to denoise noisy actions through a learned reverse process. *Diffusion Policy* (Chi et al., 2024) adopts this framework by generating short-horizon action trajectories conditioned on past observations, effectively enabling receding-horizon planning. This formulation allows for modeling complex, temporally coherent, and multi-modal behaviors, making it well-suited for high-dimensional imitation learning from raw demonstrations.

**Flow-based Behavior Cloning** offers a deterministic alternative via *flow matching*. Instead of iteratively denoising, the model learns a time-dependent velocity field $\mathbf{v}_\theta$ that maps a noise sample $\mathbf{z} \sim p_0$ to an expert action $a$ along a linear interpolation path $a_\tau = (1 - \tau)\mathbf{z} + \tau a$. The model is trained using the flow matching objective:

$$\mathcal{L}_{\text{flow-BC}} = \mathbb{E}_{(s,a)\sim\mathcal{D},\, \mathbf{z}\sim p_0,\, \tau\sim p(\tau)} \left[ \|\mathbf{v}_\theta(a_\tau, s, \tau) - (a - \mathbf{z})\|_2^2 \right]. \tag{8}$$

At inference, an action sample ($a_1$) is generated by solving the following learned ODE:

$$\frac{da_\tau}{d\tau} = \mathbf{v}_\theta(a_\tau, s, \tau), \quad a_0 \sim p_0. \tag{9}$$

## 3 SINGLE STEP COMPLETION MODEL

In this section, we begin by introducing a general framework for generative modeling using single-step completion models. Unlike conventional flow-based approaches in which the models learn to predict instantaneous velocity fields conditioned on intermediate states, our method incorporates an additional objective to learn *completion shortcuts*, normalized directions that complete the generative process in a single step from any intermediate states in the flow trajectory. This yields an expressive and efficient policy class that bypasses costly iterative generation, making it particularly suitable for imitation and reinforcement learning settings introduced earlier. We conclude the section by developing an offline RL algorithm that leverages this Single Step Completion Policy for continuous control. In the next section, we extend this completion modeling approach to derive flat inference policies that can effectively exploit subgoal structure in long-horizon goal-conditioned tasks.

### 3.1 LEARNING TO PREDICT SHORTCUTS

Unlike standard flow-matching models that predict only instantaneous velocities at $\tau$, requiring many small integration steps, **shortcut models** additionally condition on the desired step size $d$, enabling accurate long-range jumps that account for trajectory curvature and avoid discretization errors from large naive steps (Frans et al., 2024). Formally, a shortcut model learns the normalized direction $h(x_\tau, \tau, d)$ that when scaled by step size $d$ produces the correct next point: $x'_{\tau+d} = x_\tau + h(x_\tau, \tau, d) \cdot d$. Frans et al. (2024) train the parametrized model $h_\theta(x_\tau, \tau, d)$ using the following additional *self-consistency* loss:

$$\mathcal{L}_{\text{shortcut}} = \mathbb{E}_{\tau, x_0, x_1, d} \left[ \|h_\theta(x_\tau, \tau, d) - h_\theta^{\text{target}}(x_\tau, \tau, d)\|_2^2 \right], \tag{10}$$

where $h_\theta^{\text{target}}(x_\tau, \tau, d)$ is a bootstrap target predicted by the model itself, but treated as fixed using `stop-gradient` operation to block gradient flow during optimization. It is given by:

$$h_\theta^{\text{target}}(x_\tau, \tau, d) = \frac{1}{2} \left\{ h_\theta(x_\tau, \tau, d/2) + h_\theta(x'_{\tau+d/2}, \tau + d/2, d/2) \right\},$$

where $x'_{\tau+d/2} = x_\tau + h_\theta(x_\tau, \tau, d/2)d/2$.

### 3.2 LEARNING TO COMPLETE THE GENERATIVE PROCESS

While the self-consistency objective in Equation 10 enables training shortcut models without expensive ground-truth integration, it introduces a fundamental challenge: the targets used for supervision are themselves generated by the model. Consequently, these bootstrap targets are only reliable once the model has attained a certain level of accuracy, posing difficulties in early training with inaccurate predictions and in deep RL settings with evolving or implicit target policies.

Frans et al. (2024) propose a mixed-objective training scheme wherein a fraction of the training batch is supervised using standard flow-matching loss with ground-truth velocity targets, while the remainder is trained using the self-consistency loss. The proportion of flow-matching supervision is treated as a tunable hyperparameter, allowing the model to rely more heavily on stable targets during the initial training phases and gradually transition to self-supervised learning as performance improves. However, in domains such as robot learning, and in particular offline reinforcement learning, the reliance on internally generated pseudo-targets may be problematic. In these settings, generative models are often employed for behavior cloning or as part of behavior-constrained regularization to address distributional shift. If the shortcut model is trained using inaccurate self-generated targets, especially in early stages, it may lead to model drift or unsafe extrapolation. Therefore, while the self-consistency loss offers scalability and flexibility, its applicability in offline settings with distributional constraints warrants careful consideration.

**Single-step completion vector.** To integrate the shortcut learning process into a Q-learning framework and address the limitations of bootstrap targets in shortcut models, we propose an alternative approach based on *single-step completion prediction*. Specifically, at each intermediate point along the flow or transport path, the model is additionally trained to predict a *completion vector*—the normalized direction from the current sample to the final target sample (see Figure 1). This auxiliary prediction encourages the model's local dynamics to remain aligned with the overall trajectory, maintaining local consistency while enabling high-quality sample generation. In low-dimensional distribution learning problems such as action spaces, this approach facilitates the learning of highly expressive multi-modal policies compared to unimodal alternatives. Unlike bootstrap-based shortcut methods that rely on self-generated targets, this formulation helps to directly predict the remaining trajectory toward the terminal point $x_1$ via a normalized direction vector. Formally, at any intermediate timestep $\tau \in [0, 1]$, we define the single-step completion vector $h_\theta(x_\tau, \tau, d)$, with $d$ set to $(1-\tau)$, as the normalized direction from $x_\tau$ to the final state $x_1$. When scaled by the remaining time $(1-\tau)$, this vector yields the step required to transport $x_\tau$ to $x_1$:

$$\hat{x}_1 = x_\tau + h_\theta(x_\tau, \tau, 1-\tau) \cdot (1-\tau). \tag{11}$$

This design allows for direct supervision from data, as $x_1$ is drawn from the training distribution. The model is trained to minimize the squared error between the predicted final point $\hat{x}_1$ and the true $x_1$:

$$\mathcal{L}_{\text{completion}} = \mathbb{E}_{\tau, x_\tau, x_1} \left[ \|x_\tau + h_\theta(x_\tau, \tau, 1-\tau) \cdot (1-\tau) - x_1\|_2^2 \right], \tag{12}$$

where $x_\tau = (1-\tau)x_0 + \tau x_1$ is a linear interpolation between $x_0 \sim p_0$ and $x_1 \sim \mathcal{D}$. Alternatively, setting $d = 0$ in $h_\theta(x_\tau, \tau, d)$ corresponds to the instantaneous flow velocity at $(x_\tau, \tau)$.

**From completion vectors to policy representation.** By training against ground-truth final samples from the dataset, the *completion vector* prediction approach bypasses the need for bootstrap targets and mitigates the risk of model-induced distributional errors, making it particularly suitable for applications in offline and safety-critical settings. While we present a general method for generative modeling using the single-step completion models described above, this formulation can be naturally adapted to policy representation by an additional conditioning on the environment state. We term this **Single-Step Completion Policy (SSCP)**, enabling supervised policy learning via behavior cloning or related techniques in learning-based continuous control. This policy class offers an additional advantage in the context of RL, where the same action samples generated via the single-step completion vector can also be employed in computing the deterministic policy gradient loss:

$$\mathcal{L}_{\pi_Q}(\theta) = \mathbb{E}_{s \sim \mathcal{D}} \left[ -Q^{\pi_\theta}(s, \pi_\theta(s)) \right], \tag{13}$$

where $\pi_\theta(s)$ denotes the policy that uses the single-step completion model as a function approximator. The policy output is computed as:

$$\pi_\theta(s) = a_\tau + h_\theta(a_\tau, s, \tau, 1 - \tau) \cdot (1 - \tau), \qquad \tau \in [0, 1) \tag{14}$$

where $a_\tau$ is an intermediate action along the forward flow from $a_0$ to $a_1$, conditioned on the state.

Notes on nomenclature: (1) The subscript $\tau$ in $a_\tau$ denotes the interpolation timestep in the generative transport process, not the timestep along the reinforcement learning trajectory. Specifically, $a_0$ corresponds to noise, $a_1$ is the behavior cloning target, and $a_\tau$ is the interpolated representation at intermediate step $\tau$. (2) While we refer to the loss above as a deterministic policy gradient loss, the policy induced by the single-step completion model is not deterministic in a probabilistic sense, as it defines a transport between distributions. However, for a fixed input $a_\tau$, the output $\pi_\theta(s)$ is deterministic, since both the model and the generative transport process are fully deterministic.

### 3.3 OFFLINE RL WITH SINGLE STEP COMPLETION POLICY

We now instantiate our single-step completion model as an expressive and efficient policy for offline RL in continuous control, termed **SSCQL** (Single-Step Completion Q-Learning). The actor training objective in this method combines a flow-matching loss, a shortcut-based completion loss, and a Q-learning objective. The *flow loss* aligns the model's velocity field with intermediate points along the generative path:

$$\mathcal{L}_{\text{flow}}(\theta) = \mathbb{E}_{s,a\sim\mathcal{D},\, z\sim p_0,\, \tau\sim p(\tau)} \left[ \|h_\theta(a_\tau, s, \tau, 0) - (a - z)\|_2^2 \right]. \tag{15}$$

The *completion loss* directly supervises the model to regress to the terminal action in one step:

$$\mathcal{L}_{\text{completion}}(\theta) = \mathbb{E}_{s,a\sim\mathcal{D},\, z\sim p_0,\, \tau\sim p(\tau)} \left[ \|a_\tau + h_\theta(a_\tau, s, \tau, 1-\tau) \cdot (1-\tau) - a\|_2^2 \right]. \tag{16}$$

The total actor loss combines the flow-matching objective (Eq. 15), the completion loss (Eq. 16), and the policy gradient loss (Eq. 13):

$$\mathcal{L}_\pi(\theta) = \alpha_1 \mathcal{L}_{\text{flow}}(\theta) + \alpha_2 \mathcal{L}_{\text{completion}}(\theta) + \mathcal{L}_{\pi_Q}(\theta). \tag{17}$$

where $\alpha_1$ and $\alpha_2$ tune the strength of regularization that constrains the actor within the dataset (behavior policy) distribution. The critic is trained using the following Bellman backup loss:

$$\mathcal{L}_Q(\phi) = \mathbb{E}_{(s,a,r,s')\sim\mathcal{D}} \left[ \left( Q_\phi(s,a) - \left( r + \gamma \cdot \min_{i=1,2} Q_{\phi'_i}(s', \pi_\theta(s')) \right) \right)^2 \right]. \tag{18}$$

Compared to standard Flow-Matching (or Diffusion) -based policies, the shortcut completion mechanism eliminates the need for iterative generation across three stages. First, training flow-matching actors with off-policy methods like DDPG typically requires generating actions via multi-step flow rollouts and backpropagating through the entire computation chain, which is both memory- and compute-intensive. Second, critic training in standard Q-learning involves sampling actions from the current policy at the next state, i.e., $Q_{\phi'}(s', \pi_\theta(s'))$, which again demands full flow trajectory rollout during training. By enabling single-step inference, our method resolves both issues efficiently. Third, test-time action generation is similarly reduced to a single forward pass: $\pi_\theta(s) = z + h_\theta(z, s, 0, 1)$, where $z \sim p_0$. The complete algorithm is presented in Appendix A.2.

## 4 GOAL-CONDITIONED RL: DISTILLING HIERARCHY INTO FLAT POLICIES

### 4.1 OFFLINE GOAL-CONDITIONED REINFORCEMENT LEARNING

Goal-conditioned RL (GCRL) augments standard RL by conditioning the policy on a target goal $g \in \mathcal{G}$, enabling agents to reach diverse goals without retraining. In offline settings, GCRL leverages static datasets and sparse goal-reaching rewards, typically $r(s, a, g) = -\mathbb{1}(s \neq g)$, thereby avoiding the need for dense or engineered signals. The resulting goal-conditioned policy $\pi(a \mid s, g)$ must generalize across initial states and goals. Recent methods such as RIS (Chane-Sane et al., 2021), POR (Xu et al., 2022), and HIQL (Park et al., 2023) adopt a hierarchical structure that decomposes the problem into high-level subgoal selection and low-level executable action selection, significantly outperforming flat policies. HIQL (Hierarchical IQL), which we build upon and compare against, learns both a high-level policy $\pi_{\theta_h}^h(s_{t+k} \mid s_t, g)$ and a low-level policy $\pi_{\theta_\ell}^\ell(a_t \mid s_t, s_{t+k})$ via advantage-weighted regression (Peng et al., 2019; Kostrikov et al., 2021):

$$\mathcal{L}_{\pi_h}(\theta_h) = \mathbb{E}_{(s_t, s_{t+k}, g)} \left[ -\exp\left( \beta \cdot A^h(s_t, s_{t+k}, g) \right) \log \pi_{\theta_h}^h(s_{t+k} \mid s_t, g) \right],$$
$$\mathcal{L}_{\pi_\ell}(\theta_\ell) = \mathbb{E}_{(s_t, a_t, s_{t+1}, s_{t+k})} \left[ -\exp\left( \beta \cdot A^\ell(s_t, a_t, s_{t+k}) \right) \log \pi_{\theta_\ell}^\ell(a_t \mid s_t, s_{t+k}) \right].$$

### 4.2 LEARNING TO REACH GOALS VIA COMPLETION MODELING

While hierarchical decomposition, as in HIQL, can enhance generalization in offline goal-conditioned reinforcement learning (GCRL), it remains an open question whether explicit hierarchy is strictly necessary for subgoal discovery and exploitation. A promising and somewhat straightforward alternative is to distill the benefits of subgoal guidance into a non-hierarchical architecture, thereby simplifying both training and inference while retaining the representational power of hierarchical policies. Our approach draws inspiration from shortcut modeling and completion prediction

frameworks introduced in earlier sections, leveraging them to learn flat goal-conditioned policies from offline data. Specifically, we extend the shortcut-based completion framework to a flat, goal-conditioned setting by leveraging a single unified model capable of hierarchical reasoning through conditional inputs.

To exploit the subgoal structure within a flat architecture without explicit use of hierarchical learning, we propose a joint policy model $\pi_\theta$ that outputs both immediate action $a_t$ and an intermediate subgoal $s_{t+k}$, conditioned on the current state $s_t$, the final goal $g$, and an abstract hierarchy level indicator $\mathtt{k} \in \{0, 1\}$. The level variable $\mathtt{k}$ captures coarse (goal-level) or fine-grained (subgoal-level) reasoning, en-

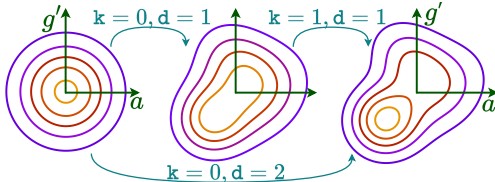

Figure 2: Hierarchy Distillation with Shortcuts

abling shared parameterization across the hierarchy: $(a_t, g') \sim \pi_\theta(\cdot \mid s_t, g, \mathtt{k})$, where $g'$ denotes either a subgoal $s_{t+k}$ (for $\mathtt{k} = 0$) or a next state $s_{t+1}$ (for $\mathtt{k} = 1$). The model is trained using advantage-weighted regression losses derived from the hierarchical Q-function, with separate losses corresponding to each level of reasoning:

$$\mathcal{L}_0(\theta) = \mathbb{E}_{(s_t, a_t, s_{t+k}, g)} \left[ -\exp\left(\beta \cdot A^h(s_t, s_{t+k}, g)\right) \log \pi_\theta(a_t, s_{t+k} \mid s_t, g, \mathtt{k} = 0) \right],$$
$$\mathcal{L}_1(\theta) = \mathbb{E}_{(s_t, a_t, s_{t+1}, s_{t+k})} \left[ -\exp\left(\beta \cdot A^\ell(s_t, a_t, s_{t+k})\right) \log \pi_\theta(a_t, s_{t+1} \mid s_t, s_{t+k}, \mathtt{k} = 1) \right].$$

At inference, hierarchical reasoning is simulated through a two-stage procedure: first, the model samples a subgoal from the high-level distribution, $(\hat{a}_t, s_{t+k}) \sim \pi_\theta(\cdot \mid s_t, g, \mathtt{k} = 0)$; then, conditioned on the predicted subgoal, a low-level action is produced via $(a_t, s_{t+1}) \sim \pi_\theta(\cdot \mid s_t, s_{t+k}, \mathtt{k} = 1)$. This is similar in spirit to flow-matching methods where same velocity function approximator is used to predict velocity across all the points in the flow trajectory.

**Shortcut-Based Flat Policy.** To enable direct flat inference through multi-step completion, we augment the model input with an additional variable $\mathtt{d}$, which specifies the number of forward steps in the hierarchy. This variable acts analogously to step size $d$ in the original shortcut framework and allows the model to simulate multi-step reasoning using different levels of abstraction within the same architecture: $(a_t, g') \sim \pi_\theta(\cdot \mid s_t, g, \mathtt{k}, \mathtt{d})$.

Training in this setting uses analogous loss functions, policy now conditioned on both $\mathtt{k}$ and $\mathtt{d}$:

$$\mathcal{L}_0(\theta) = \mathbb{E}_{(s_t, a_t, s_{t+k}, g) \sim \mathcal{D}} \left[ -\exp\left(\beta \cdot A^h(s_t, s_{t+k}, g)\right) \log \pi_\theta(a_t, s_{t+k} \mid s_t, g, \mathtt{k} = 0, \mathtt{d} = 1) \right],$$
$$\mathcal{L}_1(\theta) = \mathbb{E}_{(s_t, a_t, s_{t+1}, s_{t+k}) \sim \mathcal{D}} \left[ -\exp\left(\beta \cdot A^\ell(s_t, a_t, s_{t+k})\right) \log \pi_\theta(a_t, s_{t+1} \mid s_t, s_{t+k}, \mathtt{k} = 1, \mathtt{d} = 1) \right].$$

To enable single-step prediction during test-time inference, we incorporate a self-supervised loss that aligns shortcut predictions with composed hierarchical rollouts:

$$\mathcal{L}_s(\theta) = \mathbb{E}_{(s_t, g) \sim \mathcal{D}, (a_t, s_{t+1}) \sim \pi_{\mathtt{k}=0}^{\mathtt{d}=1} \circ \pi_{\mathtt{k}=1}^{\mathtt{d}=1}} \left[ -\log \pi_\theta(a_t, s_{t+1} \mid s_t, g, \mathtt{k} = 0, \mathtt{d} = 2) \right].$$

Specifically, for $(\mathtt{k} = 0, \mathtt{d} = 2)$, we train $\pi_\theta$ to match targets $\pi_{\mathtt{k}=0}^{\mathtt{d}=1} \circ \pi_{\mathtt{k}=1}^{\mathtt{d}=1}$, with gradient flow stopped through the target. We refer to this model as Goal-Conditioned SSCP (GC-SSCP), and use this shorthand in the results section. This training enables direct inference by sampling from $\pi_\theta(\cdot \mid s_t, g, 0, 2)$ at test time, bypassing recursive hierarchical rollout.

## 5 EXPERIMENTS AND RESULTS

We empirically evaluate SSCP and its derived variants across diverse reinforcement learning settings to demonstrate their flexibility and effectiveness. While the main results are presented in this section, additional results and details, including training curves, implementation specifics, hyper-parameters, and ablations, are provided in the appendix. Across the benchmark tasks, we compare against strong baselines and report both quantitative metrics and qualitative trends to highlight the efficiency, expressiveness, and generalizability of our approach. A detailed discussion and justification of datasets/benchmarks and baselines is also provided in the appendix.

## 5.1 OFFLINE RL

We evaluate SSCQL on the D4RL benchmark (Fu et al., 2020), focusing on continuous control tasks from the MuJoCo suite: `HalfCheetah`, `Hopper`, and `Walker2d`. We consider three dataset types per environment, `medium`, `medium-replay`, and `medium-expert`, that assess the method's performance under varying data distributions and qualities. Performance is measured using normalized scores, averaged over 10 evaluation episodes. In Table 1, we report the highest-performing method in **bold**, and highlight any method within 95% of the best with boldface as well.

Table 1: Offline Reinforcement Learning Results.

| Gym Tasks | BC | TD3+BC | CQL | IQL | Diffuser | FQL | DQL | CAC | IDQL | SSCQL |
|---|---|---|---|---|---|---|---|---|---|---|
| Halfcheetah-ME | 55.2 | 90.7 | 91.6 | 86.7 | 79.8 | **102.1** | 96.8 | 84.3 | 95.9 | **98.1±0.9** |
| Hopper-ME | 52.5 | 98.0 | **105.4** | 91.5 | **107.2** | 76.7 | **111.1** | 100.4 | **108.6** | **110.9±0.9** |
| Walker2d-ME | **107.5** | **110.1** | **108.8** | **109.6** | **108.4** | 102.6 | **110.1** | **110.4** | 112.7 | **111.1±0.1** |
| Halfcheetah-M | 42.6 | 48.3 | 44.0 | 47.4 | 44.2 | 55.6 | 51.1 | **69.1** | 51.0 | 52.3±0.5 |
| Hopper-M | 52.9 | 59.3 | 58.5 | 66.3 | 58.5 | 60.6 | 90.5 | 80.7 | 65.4 | **102.4±0.2** |
| Walker2d-M | 75.3 | **83.7** | 72.5 | 78.3 | 79.7 | 65.9 | **87.0** | 83.1 | 82.5 | 84.2±0.9 |
| Halfcheetah-MR | 36.6 | 44.6 | 45.5 | 44.2 | 42.2 | 48.3 | 47.8 | **58.7** | 45.9 | 44.4±1.0 |
| Hopper-MR | 18.1 | 60.9 | 95.0 | 94.7 | **96.8** | 50.7 | **101.3** | **99.7** | 92.1 | **101.4±0.4** |
| Walker2d-MR | 26.0 | 81.8 | 77.2 | 73.9 | 61.2 | 38.8 | **95.5** | 79.5 | 85.1 | 85.9±13.4 |
| Average | 51.9 | 75.3 | 77.6 | 77.0 | 75.3 | 66.8 | **87.9** | 85.1 | 82.1 | **87.9** |

We compare SSCQL against both unimodal Gaussian policy baselines and state-of-the-art generative policy approaches. Gaussian-based methods include Behavior Cloning (BC), TD3+BC (Fujimoto & Gu, 2021), IQL (Kostrikov et al., 2021), and CQL(Kumar et al., 2020), covering both actor and critic regularization paradigms. Generative policy baselines include Diffuser (planning-based), IDQL (Hansen-Estruch et al., 2023) (rejection sampling), DQL (Wang et al., 2022), and CAC (Ding & Jin, 2023) (both actor-regularized diffusion methods). FQL (Park et al., 2025) shares a similar motivation with our approach but applies flow matching *only* for behavior cloning and distills a single-step policy into a separate MLP; a detailed comparison is provided in Appendix A.7.

Table 2 reports compute characteristics, comparing SSCQL against generative-policy baselines trained on identical GPU resources. SSCQL demonstrates a significant advantage in training and inference efficiency, being up to 64× faster in training than DQL, a strong-performing diffusion actor baseline, and offering over an order-of-magnitude speedup in inference. Overall, SSCQL achieves consistently strong performance across all tasks while offering a dramatically streamlined training and deployment profile which highlights its practical advantages in offline reinforcement learning scenarios.

Table 2: Comparison of training and inference characteristics across generative modeling baselines.

| Method | Training Time | Inference Time (ms) | Training Steps | Denoising Steps | Training Batch |
|---|---|---|---|---|---|
| DQL | ∼8 hours | 1.27 ± 0.48 | 1M | 5 | 256 |
| CAC | ∼5 hours | 0.85 ± 0.61 | 1M | 2 | 256 |
| IDQL | ∼1 hour | 2.01 ± 1.50 | 3M | 5 | 512 |
| SSCQL | ∼16 mins | 0.27 ± 0.36 | 500K | 1 | 1024 |

## 5.2 OFFLINE TO ONLINE FINETUNING AND COMPLETE ONLINE TRAINING

We evaluate SSCQL's adaptability in both offline-to-online finetuning and fully online RL. Table 3 reports finetuning results where offline-pretrained policies are further trained with online interaction. We compare against Cal-QL Nakamoto et al. (2023) (a SOTA method designed for the regime), DQL, and CAC (strong generative policy baselines). The numbers before and after '+' in the SSCQL headings denote the offline and online training steps respectively. The final scores with the best performance are bolded, and a red arrow (→) indicates a drop that exceeds 10% of the offline score. SSCQL yields stable improvements, while DQL and CAC often degrade, underscoring the brittleness of diffusion-based policies in online adaptation.

Table 3: Offline to Online Finetuning Results

| Gym Tasks | Cal-QL | DQL | CAC | SSCQL (100K+100K) | SSCQL (250K+250K) |
|---|---|---|---|---|---|
| Halfcheetah-ME | 54→99 | 96.8→103.9 | 84.3→99.6 | 96.03±0.76→97.32±1.32 | 98.69±0.82→**110.68±0.26** |
| Hopper-ME | 69→76 | 111.1→71.7 | 100.4→65.4 | 110.76±0.81→**112.05±2.36** | 111.23±1.28→**111.27±1.13** |
| Walker2d-ME | 96→77 | 110.1→**117.0** | 110.4→101.8 | 109.81±0.11→110.37±0.19 | 110.49±0.09→**115.46±0.55** |
| Halfcheetah-M | 52→93 | 51.1→**99.6** | 69.1→**98.7** | 51.84±0.59→58.93±0.51 | 53.32±0.32→60.51±0.26 |
| Hopper-M | 89→**98** | 90.5→77.2 | 80.7→60.5 | 46.12±3.48→**98.98±8.91** | 61.92±4.62→**99.83±9.60** |
| Walker2d-M | 75→103 | 87.0→**118.3** | 83.1→108.9 | 73.88±20.43→85.42±0.84 | 84.13±1.22→85.73±0.69 |
| Halfcheetah-MR | 51→**93** | 47.8→**96.3** | 58.7→80.7 | 51.24±0.40→58.59±0.39 | 47.30±0.47→61.56±0.19 |
| Hopper-MR | 76→**110** | 101.3→68.4 | 99.7→74.6 | 73.91±34.36→96.70±16.47 | 91.75±19.67→95.32±19.82 |
| Walker2d-MR | 52→99 | 95.5→95.7 | 79.5→102.0 | 92.89±15.30→101.23±8.91 | 91.88±9.61→**113.88±0.38** |

We also examine fully online RL, with results summarized in Table 4. SSCQL significantly outperforms DQL and CAC across all Gym locomotion tasks. Notably, both DQL and CAC experience reduced performance as their number of generative steps increases (5 for DQL, 2 for CAC), suggesting that deeper backward computation graphs and longer denoising chains may hinder training stability. In contrast, SSCQL employs a single-step generative actor, avoiding backpropagation through time (BPTT), which likely contributes to more stable gradients and efficient learning dynamics. More details on offline-to-online finetuning as well as fully online training are available in Appendix A.3.

Table 4: Online Reinforcement Learning Results

| Gym Tasks | DQL | CAC | SSCQL |
|---|---|---|---|
| Halfcheetah | 43.5±4.1 | 56.1±6.9 | **63.63±9.47** |
| Hopper | 79.8±21.7 | 80.2±23.3 | **105.07±4.07** |
| Walker2d | 79.8±31.23 | 74.9±36.33 | **102.90±5.26** |
| Average | 67.7 | 70.4 | **90.53** |

## 5.3 BEHAVIOR CLONING

We evaluate SSCP on behavior cloning tasks using state-based, proficient human demonstration datasets from RoboMimic (Mandlekar et al., 2021) and the contact-rich Push-T task (Florence et al., 2022). RoboMimic consists of five robotic manipulation tasks with high-quality human demonstrations, while Push-T requires precise manipulation of a T-shaped object under variable initial conditions and complex contact dynamics.

Table 5: Behavior Cloning Results

| Tasks | DP | L-G | IBC | BET | SSCP |
|---|---|---|---|---|---|
| Lift | **1.00** | **0.96** | 0.41 | **0.96** | **1.00±0.00** |
| Can | **1.00** | 0.91 | 0.00 | 0.89 | 0.99±0.01 |
| Square | **0.89** | 0.73 | 0.00 | 0.52 | **0.90±0.03** |
| Transport | **0.84** | 0.47 | 0.00 | 0.14 | **0.86±0.08** |
| Toolhang | **0.87** | 0.31 | 0.00 | 0.20 | 0.64±0.09 |
| Push-T | 0.79 | 0.61 | **0.84** | 0.70 | **0.85±0.03** |
| Average | **0.88** | 0.68 | 0.21 | 0.57 | **0.87** |

In table 5, we follow the evaluation protocol of Chi et al. (2024), and compare against strong baselines including Diffusion Policy (DP), which uses 100 iterative denoising steps. Despite its single-step generation, SSCP achieves competitive or superior performance across all tasks, matching the performance of DP while offering significantly reduced computational overhead. Other baseline methods include LSTM-GMM Mandlekar et al. (2021) (L-G), Implicit Behavior Cloning Florence et al. (2022) (IBC), and Behavior Transformer Shafiullah et al. (2022) (BET). Full experimental details and training setups are provided in Appendix C.

## 5.4 OFFLINE GOAL-CONDITIONED RL

We evaluate our proposed Goal-Conditioned Single-Step Completion Policy (GC-SSCP) in goal-conditioned offline RL using PointMaze and AntMaze tasks from OGBench, which challenge agents with long-horizon and hierarchical reasoning. The benchmark covers 16 environment-dataset combinations: {pointmaze, antmaze} × {medium, large, giant, teleport} × {navigate, stitch}. Following OGBench protocol, we report average success rates over 6000 evaluation episodes (8 training seeds × 50 test seeds × 5 goals × 3 last evaluation epochs), as shown in Table 6. For each dataset/environment, we highlight in bold both the overall best-performing method and the best-performing method among the flat-inference baselines.

We compare against goal-conditioned baselines including GCBC (Ghosh et al., 2019; Lynch et al., 2020), GCIVL/GCIQL (Kostrikov et al., 2021; Park et al., 2023; 2024a), QRL (Wang et al., 2023), CRL (Eysenbach et al., 2022), and HIQL (Park et al., 2023). Our method, GC-SSCP, consistently achieves strong performance across tasks, notably outperforming not only other flat baselines but also the hierarchical HIQL on average. Unlike HIQL, which only transfers top-level policy outputs to a separate low-level controller, GC-SSCP employs a shared policy network across abstraction

Table 6: Offline Goal Conditioned Reinforcement Learning Results.

| Inference Type → | Flat | | | | | Hierarchical | Flat |
|---|---|---|---|---|---|---|---|
| Dataset ↓ | GCBC | GCIVL | GCIQL | QRL | CRL | HIQL | GC-SSCP |
| pointmaze-medium-navigate-v0 | 9±6 | 63±6 | 53±8 | 82±5 | 29±7 | 79±5 | **92±3** |
| pointmaze-large-navigate-v0 | 29±6 | 45±5 | 34±3 | 86±9 | 39±7 | 58±5 | **95±3** |
| pointmaze-giant-navigate-v0 | 1±2 | 0±0 | 0±0 | 68±7 | 27±10 | 46±9 | **73±7** |
| pointmaze-teleport-navigate-v0 | 25±3 | 45±3 | 24±7 | 8±4 | 24±6 | 18±4 | **50±5** |
| pointmaze-medium-stitch-v0 | 23±18 | 70±14 | 21±9 | **80±12** | 0±1 | 74±6 | 80±4 |
| pointmaze-large-stitch-v0 | 7±5 | 12±6 | 31±2 | **84±15** | 0±0 | 13±6 | 14±4 |
| pointmaze-giant-stitch-v0 | 0±0 | 0±0 | 0±0 | **50±8** | 0±0 | 0±0 | 0±0 |
| pointmaze-teleport-stitch-v0 | 31±9 | **44±2** | 25±3 | 9±5 | 4±3 | 34±4 | 35±5 |
| antmaze-medium-navigate-v0 | 29±4 | 72±8 | 71±4 | 88±3 | 95±1 | **96±1** | 94±1 |
| antmaze-large-navigate-v0 | 24±2 | 16±5 | 34±4 | 75±6 | 83±4 | **91±2** | 87±2 |
| antmaze-giant-navigate-v0 | 0±0 | 0±0 | 0±0 | 14±3 | 16±3 | **65±5** | 52±2 |
| antmaze-teleport-navigate-v0 | 26±3 | 39±3 | 35±5 | 35±5 | **53±2** | 42±3 | 42±1 |
| antmaze-medium-stitch-v0 | 45±11 | 44±6 | 29±6 | 59±7 | 53±6 | **94±1** | 84±4 |
| antmaze-large-stitch-v0 | 3±3 | 18±2 | 7±2 | 18±2 | 11±2 | **67±5** | 40±8 |
| antmaze-giant-stitch-v0 | 0±0 | 0±0 | 0±0 | 0±0 | 0±0 | 2±2 | 0±0 |
| antmaze-teleport-stitch-v0 | 31±6 | **39±3** | 17±2 | 24±5 | 31±4 | 36±2 | 33±4 |
| Average | 17.69 | 31.69 | 23.81 | 48.75 | 29.06 | 50.94 | **54.44** |

levels, enabling cross-level generalization without explicit hierarchical inference. We hypothesize this design promotes more coherent learning and sample reuse. Further implementation details and experimental insights are available in Appendix B, with extended ablation studies in B.6. We include the full pseudocode for the GC-SSCP algorithm (Algorithm 3), along with training curves across all evaluated environments. The appendix also presents ablation studies examining the effect of the AWR inverse temperature hyperparameter, the impact of using hierarchical versus flat inference, and the role of subgoal selection in policy performance.

## 6 CONCLUSION

We introduced a new policy class based on single-step completion models, enabling stable behavior cloning and actor-critic learning. Building on this, we proposed SSCQL, a policy-constrained offline actor-critic framework that combines the expressiveness of generative policies with the efficiency and stability of unimodal actor approximations. By leveraging flow-matching principles, SSCQL captures rich, multimodal action distributions while remaining compatible with off-policy optimization and single-step inference. We further extended this framework to goal-conditioned tasks through GC-SSCP, which employs shared architectures across reasoning levels to support flat inference and implicit cross-level generalization without explicit hierarchy. Empirical results across standard and goal-conditioned offline RL benchmarks and behavior cloning tasks validate SSCQL's practical benefits for robot learning, including strong performance, faster training and inference, and robust online adaptation.

Despite these promising results, several limitations remain. First, the expressiveness of single-step completion models, while effective in continuous control, may be limited in capturing highly multimodal or high-dimensional behaviors compared to multi-step generative policies. Similarly, in goal-conditioned RL, bypassing explicit hierarchical structure may hinder performance in very long-horizon tasks. Second, while typical DDPG+BC methods rely on a single behavior regularization coefficient, SSCQL introduces two coefficients ($\alpha_1$, $\alpha_2$) for flow matching and completion losses respectively. Although we provide guidance for tuning (Appendix A.4), performance can be sensitive to their choice, requiring task-specific tuning of these hyperparameters in the actor loss function. Third, SSCQL is currently designed for continuous action spaces; extending it to discrete or hybrid action spaces presents additional challenges in defining completion models. Incorporating maximum-entropy formulations may also offer a path forward, potentially broadening SSCQL's applicability to online RL and other decision-making domains as well.

## 7 ACKNOWLEDGMENT

This work was partly supported by the National Science Foundation, USA, under grant NSF CNS-2313104.

## 8 REPRODUCIBILITY STATEMENT

We have taken care to provide the necessary resources and documentation to facilitate the reproducibility of our work. All experiments are conducted on publicly available datasets, with details, references, and acknowledgments provided in the appendix. The accompanying code includes clear usage and installation instructions, as well as scripts for reproducing results. Pseudocodes, hyperparameters, design choices, and corresponding ablation studies are documented in the appendix, alongside additional results that further support our claims.

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

# A  OFFLINE RL WITH SINGLE STEP COMPLETION Q-LEARNING

## A.1  OFFLINE REINFORCEMENT LEARNING

The environment in a sequential decision-making setting such as Reinforcement Learning (RL) is typically modeled as a Markov Decision Process (MDP), defined by the tuple $(\mathcal{S}, \mathcal{A}, \mathcal{P}, \mathcal{R})$. Here, $\mathcal{S}$ denotes the state space, $\mathcal{A}$ the action space, $\mathcal{P}(s'|s, a) : \mathcal{S} \times \mathcal{A} \times \mathcal{S} \to [0, 1]$ the transition probability function, and $\mathcal{R}(s, a, s') : \mathcal{S} \times \mathcal{A} \times \mathcal{S} \to \mathbb{R}$ the reward function. The objective in RL is to learn a policy $\pi$ that maximizes the expected cumulative reward when interacting with the environment. This expected return of a policy $\pi$ from an initial state distribution $\rho_0$ is given by:

$$V^\pi(s_0) = \mathbb{E}_{\mathcal{T} \sim \pi} \left[ \sum_{t=0}^{T} \gamma^t r_t \mid s_0 \sim \rho_0 \right], \tag{19}$$

where $\mathcal{T} = (s_0, a_0, r_0, s_1, \dots)$ denotes a trajectory and $\gamma \in [0, 1]$ is a discount factor. The corresponding state-action value function and advantage function are defined as:

$$Q^\pi(s, a) = \mathbb{E}_{\mathcal{T} \sim \pi} \left[ \sum_{t=0}^{T} \gamma^t r_t \mid s_0 = s, a_0 = a \right], \qquad A^\pi(s, a) = Q^\pi(s, a) - V^\pi(s).$$

In *offline* RL, however, the agent does not have access to online interactions with the environment (Lange et al., 2012; Levine et al., 2020). Instead, it must learn solely from a fixed dataset $\mathcal{D} := \{(s, a, r, s')\}$, which is typically collected by an unknown behavior policy $\pi_\beta$. This setting presents unique algorithmic challenges, as the learning process is confined to the empirical distribution induced by $\pi_\beta$, without the ability to actively explore or query new transitions.

**Policy Regularization.** A core difficulty in offline RL arises from the distributional shift between the learned policy $\pi$ and the behavior policy $\pi_\beta$ used to collect the data. When $\pi$ selects actions that are not well-represented in the dataset, the value estimates can become inaccurate, leading to poor policy performance. This is often referred to as the problem of out-of-distribution (OOD) generalization or extrapolation error (Fujimoto et al., 2019; Kumar et al., 2020; Levine et al., 2020). To mitigate such issues, many offline RL methods restrict the learned policy to remain close to the behavior policy by enforcing explicit divergence constraints. A general formulation is:

$$\max_{\pi} V^\pi(s), \quad \text{subject to } D_{\mathrm{KL}}(\pi \| \pi_\beta) \leq \delta, \tag{20}$$

where $D_{\mathrm{KL}}$ denotes a statistical divergence measure (e.g., KL divergence), and $\delta$ is a tolerance parameter. While $\pi_\beta$ is unknown in practice, only a finite sample from its state-action visitation distribution is observed through $\mathcal{D}$.

This constrained objective motivates a family of approaches that regularize policy updates to remain within the support of the dataset. These methods aim to directly optimize the expected return while ensuring that the learned policy does not diverge significantly from the behavior policy, a strategy often referred to as *behavior-constrained policy optimization*. A common instantiation of this idea is the use of behavior cloning (BC) as a regularizer in the policy update. Representative examples include weighted behavior cloning techniques (eg. Peters & Schaal (2007); Peng et al. (2019); Nair et al. (2020); Koirala & Fleming (2024)), as well as rejection sampling strategies guided by generative modeling of the behavior policy (eg. Chen et al. (2022); Hansen-Estruch et al. (2023)). In this work, we consider an objective that combines off-policy policy gradient (like Lillicrap et al. (2015); Fujimoto et al. (2018)) with a behavior cloning loss. The objective function of such DDPG+BC style policy extraction is (Fujimoto & Gu, 2021; Wang et al., 2022):

$$\max_{\pi} \mathbb{E}_{(s,a) \sim \mathcal{D}} \left[ Q(s, \pi(s)) + \alpha \log \pi(a \mid s) \right], \tag{21}$$

The regularization coefficient $\alpha$ controls the trade-off between exploiting high-value actions and adhering to the support of the offline data. This formulation provides a flexible framework for offline policy learning and serves as the foundation for our offline RL method that we discuss in detail in the following sections.

---

**Algorithm 1** Single-Step Completion Policy Q-Learning (SSCQL)

---

**Initialize:** policy parameters $\theta$, critic parameters $\phi$
Initialize target networks: $\theta' \leftarrow \theta$, $\phi' \leftarrow \phi$
**for** $N$ gradient steps **do**

    Sample batch $\{(s, a, r, s')\} \sim \mathcal{D}$

    **Critic Update:**                                                     $\triangleright$ see Eq 26
    Compute target: $y \leftarrow r + \gamma \min_{i=1,2} Q_{\phi'_i}(s', \pi_{\theta'}(s'))$
    $\mathcal{L}_Q(\phi_1, \phi_2) = \sum_i \|Q_{\phi_i}(s, a) - y\|^2$
    $\phi_i \leftarrow \phi_i - \eta_\phi \nabla_{\phi_i} \mathcal{L}_Q(\phi_1, \phi_2)$

    **Actor Update:**                                                    $\triangleright$ see Eq. 22-25
    Sample noise $x_0 \sim \mathcal{N}(0, I)$, set $x_1 \leftarrow a$
    Sample $\tau \sim p(\tau)$ and set $x_\tau \leftarrow (1-\tau)x_0 + \tau x_1$

    *Velocity Prediction Loss:*                                            $\triangleright$ see Eq. 22
    $v \leftarrow x_1 - x_0$
    $\hat{v} = h_\theta(s, x_\tau, \tau, 0)$
    $\mathcal{L}_{\text{flow}}(\theta) = \|\hat{v} - v\|^2$

    *Completion Prediction Loss:*                                    $\triangleright$ see Eq. 23
    $\hat{h} = h_\theta(s, x_\tau, \tau, 1 - \tau)$
    $\hat{a} = x_t + \hat{h}(1 - \tau)$
    $\mathcal{L}_{\text{completion}}(\theta) = \|\hat{a} - a\|^2$

    *Q Loss:*                                                            $\triangleright$ see Eq. 24
    $\mathcal{L}_{\pi_Q}(\theta) = -\frac{1}{2}\left(Q_{\phi_1}(s, \hat{a}) + Q_{\phi_2}(s, \hat{a})\right)$

    *Total Actor Loss:*
    $\mathcal{L}_\pi(\theta) = \alpha_1 \mathcal{L}_{\text{flow}}(\theta) + \alpha_2 \mathcal{L}_{\text{completion}}(\theta) + \mathcal{L}_{\pi_Q}(\theta)$
    $\theta \leftarrow \theta - \eta_\theta \nabla_\theta \mathcal{L}_\pi(\theta)$

    **Target Updates:**
    $\phi'_i \leftarrow \mathrm{T}_\phi \phi_i + (1 - \mathrm{T}_\phi)\phi'_i$
    $\theta' \leftarrow \mathrm{T}_\theta \theta + (1 - \mathrm{T}_\theta)\theta'$

**end for**

---

## A.2 SSCQL: ALGORITHM AND OTHER DETAILS

As discussed earlier, a common formulation in offline reinforcement learning is to optimize a policy $\pi$ to maximize its expected return while constraining divergence from the behavior policy $\pi_\beta$, which generated the dataset $\mathcal{D}$. Since $\pi_\beta$ is unknown and only accessible through samples, practical approaches impose a soft constraint by augmenting the objective with a regularization term that encourages similarity to the dataset actions. This leads to methods that balance off-policy value maximization with behavior cloning, where a temperature parameter $\alpha$ controls the trade-off between exploitation and adherence to the empirical data distribution. In our method, the policy $\pi_\theta$ is implicitly induced by a completion model $h_\theta$ as described in Section 3.3, and we decompose the behavior cloning regularization into two terms: a flow-matching loss and a shortcut prediction loss. Specifically, given an initial noise $z \sim \mathcal{N}(0, I)$ and a ground-truth action $a$, we define $a_\tau = (1-\tau)z + \tau a$ as the interpolated sample at time $\tau \sim p(\tau)$.

**Flow Matching Loss.** To enforce local flow consistency, we predict the instantaneous velocity of the transport/generation at $(a_\tau, \tau)$ and minimize the deviation from the true velocity $a - z$:

$$\mathcal{L}_{\text{flow}} = \mathbb{E}_{s,a\sim\mathcal{D},\, z\sim\mathcal{N}(0,I),\, \tau\sim p(\tau)} \left[\|h_\theta(a_\tau, s, \tau, 0) - (a - z)\|_2^2\right]. \tag{22}$$

**Shortcut Completion Loss.** To enable consistency in long-range predictions and efficient action generation, we train the model to predict a normalized completion vector from $a_\tau$ to $a$ and penalize deviation from the target:

$$\mathcal{L}_{\text{completion}} = \mathbb{E}_{s,a\sim\mathcal{D},\, z\sim\mathcal{N}(0,I),\, \tau\sim p(\tau)} \left[\|a_\tau + h_\theta(a_\tau, s, \tau, 1-\tau) \cdot (1-\tau) - a\|_2^2\right]. \tag{23}$$

**Q-Guided Policy Loss.** To steer the model toward high-value completions, we add a critic-guided (off-policy policy gradient) loss based on the current estimate of the Q-function:

$$\mathcal{L}_{\pi_Q} = \mathbb{E}_{s \sim \mathcal{D}} \left[ -Q_\phi(s, \pi_\theta(s)) \right]. \tag{24}$$

**Total Actor Objective.** The total loss for the actor model $h_\theta$ combines these terms:

$$\mathcal{L}_\pi(\theta) = \alpha_1 \mathcal{L}_{\text{flow}} + \alpha_2 \mathcal{L}_{\text{completion}} + \mathcal{L}_{\pi_Q}, \tag{25}$$

where $\alpha_1$ and $\alpha_2$ are scalar weights that balance the contribution of the flow and completion losses, respectively, and are treated as tunable hyperparameters.

**Critic Update.** We learn the critic $Q_\phi$ using temporal-difference learning with target networks $\phi'$:

$$\mathcal{L}_Q(\phi) = \mathbb{E}_{(s,a,r,s') \sim \mathcal{D}} \left[ \left( Q_\phi(s, a) - \left( r + \gamma \cdot \min_{i=1,2} Q_{\phi_i'}(s', \pi_\theta(s')) \right) \right)^2 \right]. \tag{26}$$

**Inference.** At test time, actions are sampled by drawing a noise vector $x_0 \sim \mathcal{N}(0, I)$ and applying the one-step completion policy:

$$a = \pi_\theta(s) := x_0 + h_\theta(s, x_0, \tau = 0, d = 1), \tag{27}$$

as detailed in Algorithm 2.

---

**Algorithm 2** Sampling from the Single-Step Completion Policy $\pi_\theta$

---

**procedure** $\pi_\theta(s)$                                                  ▷ Induced policy via completion prediction
        $x_0 \sim \mathcal{N}(0, I)$
        $\tau \leftarrow 0, \quad d \leftarrow 1$
        $\hat{v} = h_\theta(s, x_0, \tau, d)$                                            ▷ see Eq. 27)
    **return**     $x_0 + \hat{v} \cdot d$
**end procedure**

---

### A.3    Offline to Online Finetuning and Complete Online Training Results

We evaluate SSCQL's adaptability in both offline-to-online finetuning and fully online RL. Online finetuning and online reinforcement learning using our proposed policy class follow a straightforward extension of the offline RL algorithm presented in algorithms 1 and 2. The key difference in online finetuning lies in the transition from purely offline training to incorporating online interactions: after a fixed number of offline training steps, the replay buffer begins to accumulate new transitions from agent-environment interactions. In contrast, in the fully online RL setting, the replay buffer is initialized as empty and is populated exclusively with data collected during training, without relying on the offline D4RL dataset. For offline-to-online finetuning, we report performance across varying numbers of offline training and finetuning steps. In the online RL experiments, agents are trained for 2M gradient steps. Unless otherwise specified, the network architecture and implementation details remain consistent with those used in the offline RL setting. However, the flow matching loss coefficient and the completion loss coefficient were separately tuned for these experiments; their values are reported in Table 10.

Table 7 reports finetuning results where offline-pretrained policies are further trained with online interaction. We compare against Cal-QL Nakamoto et al. (2023) (a SOTA method designed for the regime), DQL, and CAC (strong generative policy baselines). Final scores with the best performance are bolded, and a red arrow ($\rightarrow$) indicates a drop exceeding 10% from the offline score. SSCQL yields stable improvements, while DQL and CAC often degrade, underscoring the brittleness of diffusion-based policies in online adaptation.

Table 7: Offline to Online Finetuning Results

| Gym Tasks | Cal-QL | DQL | CAC | SSCQL (100K+100K) | SSCQL (250K+250K) |
|---|---|---|---|---|---|
| Halfcheetah-ME | 54→99 | 96.8→103.9 | 84.3→99.6 | 96.03±0.76→97.32±1.32 | 98.69±0.82→**110.68±0.26** |
| Hopper-ME | 69→76 | 111.1→71.7 | 100.4→65.4 | 110.76±0.81→**112.05±2.36** | 111.23±1.28→**111.27±1.13** |
| Walker2d-ME | 96→77 | 110.1→**117.0** | 110.4→101.8 | 109.81±0.11→110.37±0.19 | 110.49±0.09→**115.46±0.55** |
| Halfcheetah-M | 52→93 | 51.1→**99.6** | 69.1→**98.7** | 51.84±0.59→58.93±0.51 | 53.32±0.32→60.51±0.26 |
| Hopper-M | 89→**98** | 90.5→77.2 | 80.7→60.5 | 46.12±3.48→**98.98±8.91** | 61.92±4.62→**99.83±9.60** |
| Walker2d-M | 75→103 | 87.0→**118.3** | 83.1→108.9 | 73.88±20.43→85.42±0.84 | 84.13±1.22→85.73±0.69 |
| Halfcheetah-MR | 51→**93** | 47.8→**96.3** | 58.7→80.7 | 51.24±0.40→58.59±0.39 | 47.30±0.47→61.56±0.19 |
| Hopper-MR | 76→**110** | 101.3→68.4 | 99.7→74.6 | 73.91±34.36→96.70±16.47 | 91.75±19.67→95.32±19.82 |
| Walker2d-MR | 52→99 | 95.5→95.7 | 79.5→102.0 | 92.89±15.30→101.23±8.91 | 91.88±9.61→**113.88±0.38** |

We also examine fully online RL (2M training steps, 3 random training seeds), with results summarized in Table 8. SSCQL significantly outperforms DQL and CAC across all Gym locomotion tasks. Notably, both DQL and CAC experience reduced performance as their num-

Table 8: Online Reinforcement Learning Results

| Gym Tasks | DQL | CAC | SSCQL |
|---|---|---|---|
| Halfcheetah | 43.5±4.1 | 56.1±6.9 | **63.63±9.47** |
| Hopper | 79.8±21.7 | 80.2±23.3 | **105.07±4.07** |
| Walker2d | 79.8±31.23 | 74.9±36.33 | **102.90±5.26** |
| Average | 67.7 | 70.4 | **90.53** |

ber of generative steps increases (5 for DQL, 2 for CAC), suggesting that deeper backward computation graphs and longer denoising chains may hinder training stability. In contrast, SSCQL employs a single-step generative actor, avoiding backpropagation through time (BPTT), which likely contributes to more stable gradients and efficient learning dynamics.

## A.4 IMPLEMENTATION DETAILS AND HYPERPARAMETERS

The common hyperparameters used across all tasks are listed in Table 9, reducing the need for extensive hyperparameter tuning. In this setup, tuning was limited primarily to the coefficients $\alpha_1$ and $\alpha_2$, which control the trade-off between behavior cloning and Q-learning. Specifically, $\alpha_1$ corresponds to the flow loss coefficient and $\alpha_2$ to the completion loss coefficient. In practice, fixing $\alpha_1 = 1.0$ and tuning only $\alpha_2$ was sufficient to obtain the results reported in Table 1. The task-specific values of $\alpha_2$ used in our experiments are provided in Table 10.

Table 9: Common Hyperparameters Across All Offline RL Experiments

| Hyperparameter | Value |
|---|---|
| Discount Factor ($\gamma$) | 0.99 |
| Batch Size | 1024 |
| Soft Update Rate for Q-Networks ($T_\phi$) | 0.005 |
| Soft Update Rate for Actors ($T_\theta$) | 0.0005 |
| Learning Rates for All Parameters | $3 \times 10^{-4}$ |
| Time Dimension | 128 |
| Training Steps | 500k |
| Actor Network Size | (512, 512, 512, 512) |
| Critic Network Size | (512, 512, 512, 512) |

In the *online reinforcement learning experiments* (sec. A.3, table 8), agents are trained for 2 million gradient steps using reduced batch sizes and smaller network architectures. Specifically, a batch size of 512 is employed, and both the actor and critic networks utilize hidden dimensions of (256, 256, 256). A smaller learning rate of $3 \times 10^{-5}$ is used consistently across experiments. Furthermore, the flow-time embedding dimension ( `time_dim`) is halved to 64. Unlike in the offline RL experiments, a gradually decreasing exploration probability is introduced during training, enabling the agent to sample random actions with higher frequency in early stages and reduce exploration over time.

**Time and step size encoding.** To provide temporal context in the flow process, we incorporate learnable Fourier feature projections. Scalar time inputs $\tau$ and step sizes $d$ are both separately encoded using a shared `FourierFeatures` module that maps each input $x$ to $[\cos(2\pi xW), \sin(2\pi xW)]$, where $W \in \mathbb{R}^{\texttt{time\_dim}/2 \times 1}$ are learnable frequency weights. The resulting embeddings are passed through separate two-layer MLPs with hidden size `time_dim`, and

Mish activation to produce temporal features $\Phi(t), \Phi(s) \in \mathbb{R}^{\texttt{time\_dim}}$. These features are combined additively and concatenated with the state and intermediate action pair $[s, a_\tau]$ as input to the final vector field network.

**Time distribution and sampling ($\mathrm{p}(\tau)$).**  To emphasize early stages of the flow process during training, we employ a non-uniform time sampling scheme. Rather than drawing time coordinates $t \sim \mathcal{U}(0, 1)$ as is common in standard flow matching, we apply the transformation $\tau = t^2$, resulting in samples from a $\mathrm{Beta}(\frac{1}{2}, 1)$ distribution. This follows from a change-of-variables analysis: if $X \sim \mathcal{U}(0, 1)$ and $Y = X^2$, then $p_Y(y) = \frac{1}{2\sqrt{y}}$, which corresponds to $\mathrm{Beta}(\frac{1}{2}, 1)$. This distribution concentrates density near zero, with approximately $32\%$ of samples falling in $[0, 0.1]$, a threefold increase over uniform sampling. Such emphasis benefits training by allocating more capacity to early-time dynamics where transitions from noise to data are most critical. Moreover, this approach remains easy to implement via squaring and aligns well with inference conditions where we evaluate at $t = 0$ in our single step completion model.

**Actor Q Loss Normalization.**  We implement an adaptive scaling for Q-based policy updates to stabilize gradient magnitudes. For each batch, we define the loss as the negative mean Q-value, scaled by the inverse of its average absolute value:

$$\mathcal{L}_{\mathrm{Q}} = \frac{1}{\mathbb{E}[|Q_\phi(s, \pi_\theta(s))|]} \cdot \left(-Q_\phi(s, \pi_\theta(s))\right)$$

This normalization ensures that the policy gradient magnitudes remain invariant to the absolute scale of Q-values and helps minimize tuning efforts for behavior cloning coefficients $\alpha_1$ and $\alpha_2$.

**Implementation.**  Our implementation is based on the JAX/Flax framework and draws inspiration from the following public repositories: `fql`[1], `IDQL`[2], `jaxrl_m`[3], and `jaxrl`[4] (Park et al., 2025; Hansen-Estruch et al., 2023; Kostrikov, 2021; Bradbury et al., 2018). Experiments were conducted on a machine with an Intel Core i9-13900KF CPU and an NVIDIA RTX 4090 GPU.

Table 10: Hyperparameter $\alpha_1$ and $\alpha_2$ used across different environments and dataset types.

| Experiments $\rightarrow$ | Offline | | Offline-to-Online | | Online | |
|---|---|---|---|---|---|---|
| Task / Dataset $\downarrow$ | $\alpha_1$ | $\alpha_2$ | $\alpha_1$ | $\alpha_2$ | $\alpha_1$ | $\alpha_2$ |
| HalfCheetah-Medium-Expert | 1.0 | 0.1 | 0.1 | 0.1 | | |
| HalfCheetah-Medium | 1.0 | 0.20 | 0.20 | 0.20 | 0.05 | 0.05 |
| HalfCheetah-Medium-Replay | 1.0 | 0.05 | 0.05 | 0.05 | | |
| Hopper-Medium-Expert | 1.0 | 0.75 | 1.0 | 0.75 | | |
| Hopper-Medium | 1.0 | 0.05 | 0.05 | 0.05 | 0.05 | 0.05 |
| Hopper-Medium-Replay | 1.0 | 0.10 | 1.0 | 0.10 | | |
| Walker2d-Medium-Expert | 1.0 | 0.1 | 0.1 | 0.1 | | |
| Walker2d-Medium | 1.0 | 0.5 | 0.5 | 0.5 | 0.05 | 0.05 |
| Walker2d-Medium-Replay | 1.0 | 0.1 | 1.0 | 0.1 | | |

## A.5   TRAINING CURVES

Figure 3 shows offline RL performance using D4RL datasets in Gym MuJoCo locomotion tasks. Figure 4 and Figure 5 present offline-to-online fine-tuning curves with 100k and 250k offline/online steps, respectively. Finally, Figure 6 shows online RL training performance in Gym MuJoCo locomotion tasks. Subplot (a) corresponds to the fully online setting with an empty replay buffer, while subplot (b) initializes the buffer with a medium-level dataset, similar to offline-to-online fine-tuning with zero offline steps (i.e., offline data is used without offline policy improvement; 0K+500K offline-to-online).

---

[1] https://github.com/seohongpark/fql
[2] https://github.com/philippe-eecs/IDQL
[3] https://github.com/dibyaghosh/jaxrl_m
[4] https://github.com/ikostrikov/jaxrl

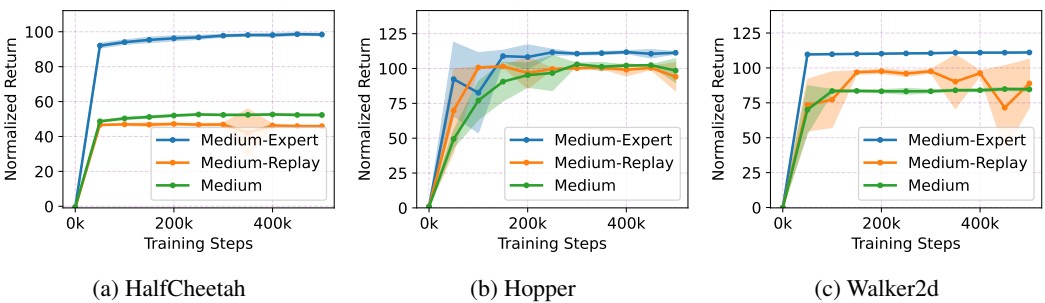

Figure 3: Training Curves for Offline RL with D4RL Datasets in Gym Mujoco Locomotion tasks

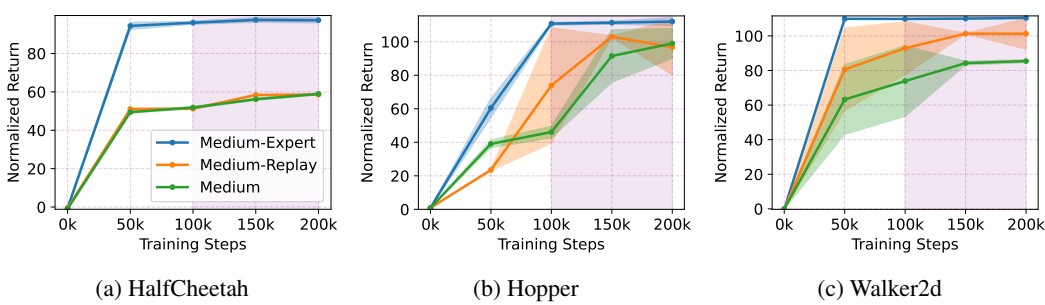

Figure 4: Training Curves for Offline to Online Finetuning with D4RL Datasets in Gym Mujoco Locomotion tasks (100k offline and 100k online steps)

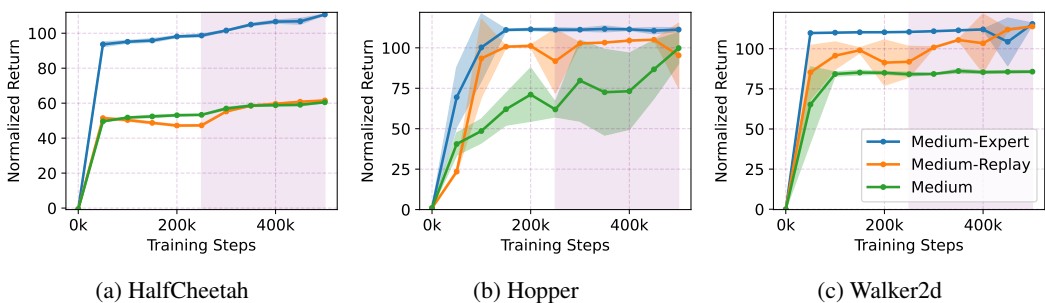

Figure 5: Training Curves for Offline to Online Finetuning with D4RL Datasets in Gym Mujoco Locomotion tasks (250k offline and 250k online steps)

## A.6 ABLATION

**Effect of the Coefficients $\alpha_1$ and $\alpha_2$ in Actor Loss.** We perform an ablation study over the behavior cloning regularization coefficients $\alpha_1$ (flow-matching loss) and $\alpha_2$ (completion loss), with $\alpha_1$ varied along the vertical axis and $\alpha_2$ along the horizontal axis in Figure 7. The column corresponding to $\alpha_2 = 0$ reflects settings where only the flow-matching loss is used, effectively removing the shortcut-completion objective. As evident across most environments, except HalfCheetah-Medium, omitting the completion loss significantly degrades one-shot action generation performance, highlighting its importance. The best results are achieved when both $\alpha_1$ and $\alpha_2$ are tuned to appropriately balance local flow consistency (via flow matching) and accurate flow trajectory completion (via completion loss). This confirms the complementary nature of the two losses: flow matching ensures smooth interpolation along the flow, while the completion loss grounds the model in final-state consistency, enabling expressive and stable behavior in offline reinforcement learning.

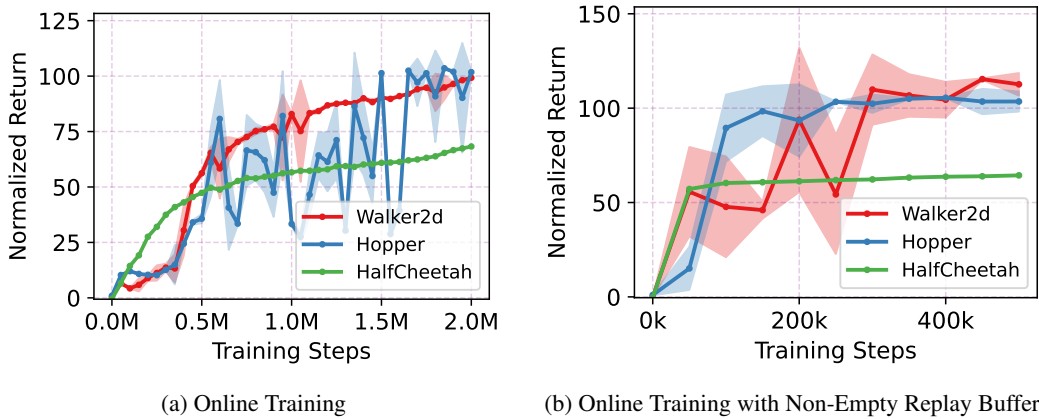

(a) Online Training                (b) Online Training with Non-Empty Replay Buffer

Figure 6: Training Curves for Online RL in Gym Mujoco Locomotion tasks. (a. Replay buffer is initialized empty. b. Replay buffer is initialized with medium-level dataset.)

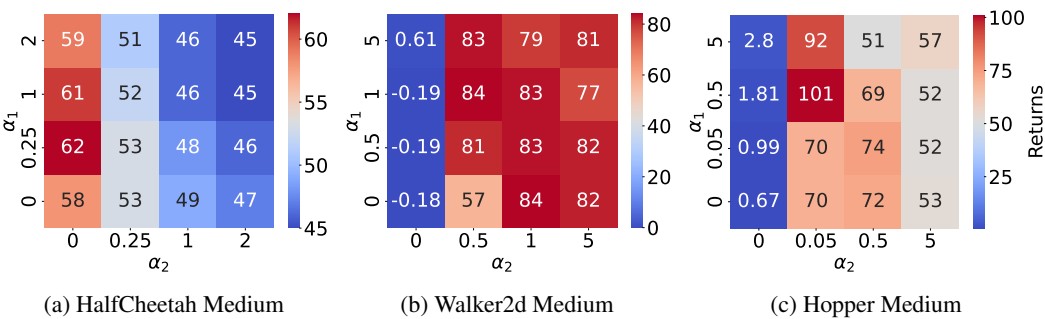

(a) HalfCheetah Medium          (b) Walker2d Medium          (c) Hopper Medium

Figure 7: Ablation on $\alpha_1$ and $\alpha_2$ for Offline RL tasks

**Effect of Bootstrap Targets in Actor Loss.**  Figure 8 evaluates the effect of replacing the completion loss with a self-supervised shortcut loss that bootstraps targets from the actor itself. The completion loss in equation 12 uses a sample (or action) from the dataset itself as a regression target, but the bootsrap shortcut loss in equation 10 uses smaller step sizes to create a *self-consistency* target. The purpose of these auxiliary losses in offline RL is to serve as a proxy to behavior cloning-based policy regularization and restrict the actor toward the behavior policy $\pi_\beta$, thereby reducing extrapolation error by discouraging out-of-distribution (OOD) actions passed to the critic. However, while the completion loss uses ground-truth actions and encourages alignment with the dataset distribution, the bootstrap shortcut loss does not provide such constraints, as its targets are generated by the model and may themselves be OOD.

As hypothesized, increasing the coefficient $\alpha_2$ of the bootstrap loss degrades performance, with larger values causing severe instability. This is especially evident in HalfCheetah-medium, where $\alpha_2 = 0$ already yields strong performance (as seen in earlier ablations), providing a clean setting to isolate the impact of the bootstrap loss. In contrast, scaling the completion loss does not drastically harm performance (Figure 7); theoretically, it biases the policy toward behavior cloning but still remains stable. In the bootstrap case, however, higher $\alpha_2$ places greater weight on poorly calibrated self-generated targets, destabilizing Q-learning as seen in the training log presented in Figure 8b.

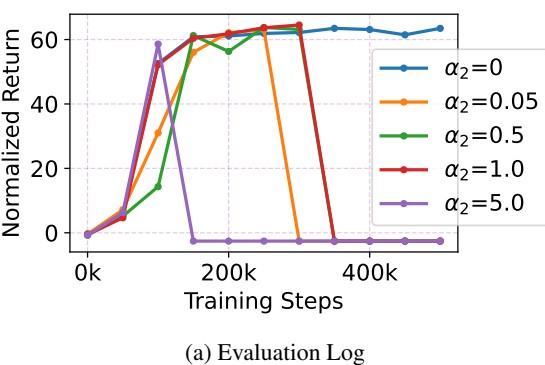
(a) Evaluation Log

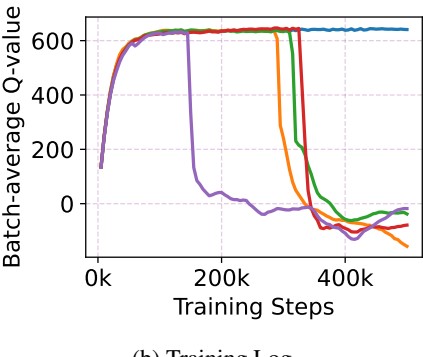
(b) Training Log

Figure 8: Replacing Completion Loss with Bootstrap Shortcut Loss in the Actor Objective (HalfCheetah Medium; $\alpha_1 = 1.0$)

## A.7 COMPARISON WITH FQL

Flow Q-Learning (FQL; Park et al. (2025)) and our method, SSCQL, both investigate single-step policy representations, but they adopt different design choices. In FQL, a flow-matching model is trained with a behavior cloning (BC) objective, and its outputs are used to regularize a separate MLP-based policy via behavior-regularized policy optimization. Importantly, the flow model is not used directly for action selection, neither during policy evaluation nor at runtime inference. In contrast, SSCQL trains a single flow-based model that serves directly as the policy. The actor and behavior regularization components both depend solely on this model.

The key distinctions are presented in the following points.

- SSCQL uses the same generative model for both training and inference. The single-step completion policy is distilled in the same flow model. However, FQL trains a separate MLP actor for policy distillation.

- SSCQL uses dataset actions directly for behavior regularization, without relying on surrogate targets from another model.

- More importantly, there are no full-flow rollouts in our training. While FQL avoids BPTT during policy optimization, it still requires full flow trajectory rollouts to generate BC targets. SSCQL avoids this entirely and remains efficient throughout.

- The final policy in SSCQL is a generative model representing the completion vector field, enabling both efficient single-step inference and optional full-flow rollout when additional compute is available; results for this variant are reported later in the section.

For empirical comparisons, Flow-QL does not report results on D4RL locomotion benchmarks. We ran the experiments ourselves and followed the suggested hyperparameter tuning, training with alpha values (0.1, 0.3, 1.0, 2.0). The results, along with the max performance (tuned hyperparameter) among the collected results, are reported below.

Table 11: Offline RL Empirical Result Comparison with Hyperparameter-Tuned FQL

| Gym Tasks | FQL($\alpha = 0.1$) | FQL($\alpha = 0.3$) | FQL($\alpha = 1.0$) | FQL($\alpha = 3.0$) | FQL(Max) | SSCQL |
|---|---|---|---|---|---|---|
| Halfcheetah-ME | 102.11 | 96.19 | 91.47 | 89.29 | **102.11** | 98.1 |
| Hopper-ME | 49.61 | 60.24 | 76.70 | 61.93 | 76.70 | **110.9** |
| Walker2d-ME | 99.11 | 97.52 | 102.57 | 92.52 | 102.57 | **111.1** |
| Halfcheetah-M | 55.63 | 52.81 | 46.87 | 43.53 | **55.63** | 52.3 |
| Hopper-M | 44.89 | 60.62 | 54.06 | 48.91 | 60.62 | **102.4** |
| Walker2d-M | 31.55 | 45.55 | 64.24 | 65.91 | 65.91 | **84.2** |
| Halfcheetah-MR | 48.33 | 44.56 | 39.43 | 39.27 | **48.33** | 44.4 |
| Hopper-MR | 27.00 | 30.54 | 50.70 | 44.95 | 50.70 | **101.4** |
| Walker2d-MR | 16.59 | 13.31 | 32.29 | 38.82 | 38.82 | **85.9** |

Unlike FQL, the final distilled policy in SSCQL remains a generative model that parameterizes a vector field. While our main focus is on single-step inference, we also implement a variant for offline

RL that performs multi-step integration along the learned completion vector field from timestep ($\tau$) 0 to 1, analogous to flow matching. Specifically, Euler's method with 10 discrete steps is used. And no additional hyperparameter tuning is performed; i.e., the same completion vector field trained for single-step generation is directly reused for the *Completion Flow* results reported in Table 12.

Table 12: Evaluation of Completion Vector Field: Multi-step (10-step) Rollout vs. Single-step Prediction. The model trained for SSCQL using single-step completion is evaluated in a 10-step rollout setting, referred to as *completion flow*.

| Environment | Completion Flow | Single Step Completion |
|---|---|---|
| Halfcheetah-M | **53.7** | 52.3 |
| Halfcheetah-MR | **47.1** | 44.4 |
| Hopper-M | **103.1** | 102.4 |
| Hopper-MR | 94.8 | **101.4** |
| Walker2d-M | 83.9 | **84.2** |
| Walker2d-MR | **98.0** | 85.9 |

These results suggest that the learned completion vector encodes a meaningful flow structure capable of guiding iterative action generation toward high-reward regions, even without explicit multi-step backpropagation during training. This highlights its dual capability: enabling efficient single-step inference by default, while also supporting flow-based rollout when additional compute is available.

## B  EXPLOITING SUBGOAL STRUCTURE IN GCRL WITH COMPLETION MODELING

### B.1  OFFLINE GOAL-CONDITIONED REINFORCEMENT LEARNING

Goal-conditioned reinforcement learning (GCRL) is a paradigm for learning general-purpose policies capable of reaching arbitrary target states without requiring policy retraining. Unlike standard reinforcement learning, where the policy is conditioned only on the current state, GCRL augments the Markov Decision Process (MDP) with a goal space $\mathcal{G}$, enabling goal-directed behavior. The goal space $\mathcal{G}$ often coincides with the state space $\mathcal{S}$ or represent task-relevant subsets of state features (e.g., spatial positions in navigation or object locations in manipulation) (Liu et al., 2022a; Andrychowicz et al., 2017).

In the offline setting, goal-conditioned RL seeks to learn from static datasets using sparse goal-matching rewards, eliminating the need for dense or hand-engineered reward functions. The objective is to learn a goal-conditioned policy $\pi(a \mid s, g) : \mathcal{S} \times \mathcal{G} \to \mathcal{A}$ that reliably reaches a specified goal state $g \in \mathcal{G}$ from arbitrary initial states. The reward function typically takes a sparse form $r(s, a, g) = -\mathbb{1}(s \neq g)$, penalizing the agent at every step until the specified goal is achieved (Park et al., 2024a; Ma et al., 2022; Sikchi et al., 2023; Xu et al., 2022).

Recent goal-conditioned RL (GCRL) methods such as RIS (Chane-Sane et al., 2021), POR (Xu et al., 2022), and HIQL (Park et al., 2023) adopt a hierarchical formulation, decomposing control into high-level subgoal selection and low-level action execution. These methods have shown strong performance over flat policies. HIQL, which shares methodological similarities with POR, learns multi-goal policies from a shared goal-conditioned value function trained via implicit Q-learning. In this work, we start with and finally compare against this line of hierarchical policy design. Specifically, HIQL introduces two policies: a high-level policy $\pi_{\theta_h}^h$ that selects subgoals, and a low-level policy $\pi_{\theta_\ell}^\ell$ that generates actions conditioned to reach those subgoals. More specifically, at test time, the high-level policy samples an intermediate subgoal $s_{t+k} \sim \pi_{\theta_h}^h(\cdot \mid s_t, g)$, which is passed to the low-level policy. The low-level policy then selects an executable action conditioned on the current state and subgoal: $a_t \sim \pi_{\theta_\ell}^\ell(\cdot \mid s_t, s_{t+k})$.

### B.2  GOAL-CONDITIONED IMPLICIT VALUE LEARNING (GCIVL).

Implicit Q-Learning (IQL) uses an asymmetric loss function in a SARSA-style policy evaluation framework to train a state-value network $V_\phi$, mitigating the impact of out-of-distribution actions during critic training (Kostrikov et al., 2021). The critic networks are trained via a TD target and an

expectile regression loss:

$$\mathcal{L}_Q(\phi_Q) = \mathbb{E}_{(s,a,s')\sim\mathcal{D}} \left[ \left( r + \gamma V_{\phi_V}(s') - Q_{\phi_Q}(s,a) \right)^2 \right] \tag{28}$$

$$\mathcal{L}_V(\phi_V) = \mathbb{E}_{(s,a)\sim\mathcal{D}} \left[ L_\xi^2 \left( Q_{\phi_Q}(s,a) - V_{\phi_V}(s) \right) \right], \tag{29}$$

where the expectile loss is defined as $L_\xi^2(u) = |\xi - \mathbf{1}(u < 0)|u^2$, with $\xi \in (0.5, 1.0)$.

We use an adaptation of IQL to an action-free variant called IVL in goal-conditioned setting to learn a *Goal-Conditioned Implicit Value (GCIVL)* (Xu et al., 2022; Park et al., 2023). Unlike standard IQL, GCIVL avoids Q-function training and instead constructs a conservative value target directly using a bootstrapped estimate from a target value network:

$$\hat{q}(s,g) = r(s,g) + \gamma V_{\phi_V'}(s',g), \tag{30}$$

where $V_{\phi_V'}$ represents target network for the goal conditioned value prediction.

The GCIVL objective then becomes:

$$\mathcal{L}_V(\phi_V) = \mathbb{E}_{(s,s',g)\sim\mathcal{D}} \left[ L_\xi^2 \left( \hat{q}(s,g) - V_{\phi_V}(s,g) \right) \right]. \tag{31}$$

To further stabilize training and reduce overestimation bias, we employ two value networks and compute the final loss as the sum of expectile losses across both.

### B.3 Hierarchical Policy Extraction, Distillation and Execution

Recent works in goal-conditioned reinforcement learning (GCRL), such as RIS (Chane-Sane et al., 2021) and HIQL (Park et al., 2023), propose hierarchical architectures that decouple the learning of complex tasks into high-level planning and low-level control. These methods employ a high-level policy to generate subgoals and a low-level policy to produce primitive actions for reaching them. Such hierarchical decomposition facilitates more structured exploration and reduces overall policy approximation error, particularly when paired with a goal-conditioned value function that serves as a consistent learning signal for both levels.

HIQL builds upon Implicit Q-Learning (IQL) and its action-free variant to train a goal-conditioned value function, and extracts both high- and low-level policies using advantage-weighted regression (AWR) objectives. Notably, the hierarchical extraction procedure in HIQL is agnostic to the specific offline RL method used to obtain the value function, as also evidenced by RIS, which employs a different critic training paradigm. The high-level policy $\pi_{\theta_h}^h$ proposes subgoals based on the current state and final goal, while the low-level policy $\pi_{\theta_\ell}^\ell$ generates actions to achieve the proposed subgoals. Both policies are optimized via AWR with their respective advantage estimators:

$$\mathcal{L}_{\pi_h}(\theta_h) = \mathbb{E}_{(s_t,s_{t+k},g)\sim\mathcal{D}} \left[ -\exp\left(\beta \cdot A^h(s_t, s_{t+k}, g)\right) \log \pi_{\theta_h}^h(s_{t+k} \mid s_t, g) \right], \tag{32}$$

$$\mathcal{L}_{\pi_\ell}(\theta_\ell) = \mathbb{E}_{(s_t,a_t,s_{t+1},s_{t+k})\sim\mathcal{D}} \left[ -\exp\left(\beta \cdot A^\ell(s_t, a_t, s_{t+k})\right) \log \pi_{\theta_\ell}^\ell(a_t \mid s_t, s_{t+k}) \right]. \tag{33}$$

At deployment time, the hierarchical agent executes as follows: given a state $s_t$ and a goal $g$, the high-level policy samples an intermediate subgoal $s_{t+k} \sim \pi_{\theta_h}^h(\cdot \mid s_t, g)$, which is provided as input to the low-level policy. The low-level policy then chooses an action conditioned on the current state and subgoal: $a_t \sim \pi_{\theta_\ell}^\ell(\cdot \mid s_t, s_{t+k})$. Our approach also follows this general philosophy: we train a goal-conditioned value function (e.g., using GCIVL) and use it to distill hierarchical properties into flat policies in a post hoc manner by exploiting subgoal structure. Loss functions for training shortcut-based flat policies to exploit this structure are:

$$\mathcal{L}_0(\theta) = \mathbb{E}_{(s_t,a_t,g',g)\sim\mathcal{D}} \left[ -\exp\left(\beta \cdot A^h(s_t, g', g)\right) \log \pi_\theta(a_t, g' \mid s_t, g, \mathbf{k}=0, \mathbf{d}=1) \right], \tag{34}$$

$$\mathcal{L}_1(\theta) = \mathbb{E}_{(s_t,a_t,s_{t+1},g')\sim\mathcal{D}} \left[ -\exp\left(\beta \cdot A^\ell(s_t, a_t, g')\right) \log \pi_\theta(a_t, s_{t+1} \mid s_t, g', \mathbf{k}=1, \mathbf{d}=1) \right], \tag{35}$$

$$\mathcal{L}_s(\theta) = \mathbb{E}_{(s_t,g)\sim\mathcal{D},(a_t,s_{t+1})\sim\pi_{\mathbf{k}=0}^{\mathbf{d}=1}\circ\pi_{\mathbf{k}=1}^{\mathbf{d}=1}} \left[ -\log \pi_\theta(a_t, s_{t+1} \mid s_t, g, \mathbf{k}=0, \mathbf{d}=2) \right]. \tag{36}$$

Here, $g'$ denotes the subgoal located $\tilde{k}$ timesteps ahead of $s_t$ in the trajectory, where $\tilde{k} = \min(k, t_g - t, T - t)$. This formulation ensures that $g'$ remains a valid subgoal, with respect to the trajectory termination at the timestep $T$, the designated goal timestep $t_g$, and the intended subgoal horizon $k$.

The first loss, $\mathcal{L}_0$ in equation 34, encourages the policy to directly optimize over intermediate states (subgoals) in the trajectory using high-level advantage estimates. In equation 35, $\mathcal{L}_1$ corresponds to standard short-horizon behavior cloning weighted by low-level advantages (AWR). Finally, in equation 36, we incorporate a shortcut loss $\mathcal{L}_s$ that distills the hierarchical decision-making process into a flat policy via supervised learning. This involves querying the hierarchical inference module (i.e., composition of $\pi_{k=0}^{d=1}$ and $\pi_{k=1}^{d=1}$) to generate behavior, which is then used as a target distribution for training the flat policy. The gradient is stopped through the target $(a_t, s_{t+1})$ used in Eq. 36. This approach enables our flat policy to inherit hierarchical structure and long-horizon reasoning capabilities without explicitly learning multiple neural network controllers (since a single shared architecture is employed) and without relying on multiple inference steps through the learned controller(s).

## B.4 Implementation Details

### B.4.1 Advantage Estimates and Goal Representation

We adopt the simplified advantage estimates introduced in Park et al. (2023) to compute the advantage-weighted regression (AWR) losses in Equations 34 and 35, leveraging a critic trained with the goal-conditioned implicit value learning (GCIVL) framework. Building on the work of Xu et al. (2022), Park et al. (2023) proposed an efficient approximation to the AWR advantage terms by replacing bootstrapped multi-step returns with value function differences. This simplification yields empirically strong results while avoiding additional critic complexity. We adopt these approximations and compute the high-level and low-level advantage estimates as:

$$\hat{A}^h(s_t,\ g',\ g) = V_{\phi_V}(g',\ g) - V_{\phi_V}(s_t,\ g), \tag{37}$$

$$\hat{A}^\ell(s_t,\ a_t,\ g') = V_{\phi_V}(s_{t+1},\ g') - V_{\phi_V}(s_t,\ g'). \tag{38}$$

where $V_{\phi_V}(\cdot, \cdot)$ denotes the GCIVL-trained value function conditioned on a goal, and the advantage estimates $\hat{A}$, computed as value differences, serve as practical proxies for true advantages in the hierarchical setting.

While our policy $\pi_\theta$ models the joint distribution over $(a_t, g')$, it does not predict the raw subgoal $g'$ (e.g., $s_{t+k}$) directly. Instead, it outputs an encoded representation $e(s, g')$ of the subgoal, where $e$ denotes a learned goal encoder. This encoder is jointly trained with the value network, such that the goal-conditioned value function is represented as $V_{\phi_V}(s, e(s, g'))$.

### B.4.2 Hierarchy Condition Representation

Our goal conditioned policy $\pi_\theta$ is additionally conditioned on discrete hierarchy indicators $\mathtt{k}$ and $\mathtt{d}$, which represent the current layer index and the step depth within the hierarchy, respectively. These indicators modulate the behavior of the shared policy network to emulate distinct roles across the hierarchy. For example, $\pi_{k=0}^{d=1}$ represents the high-level policy, responsible for subgoal inference (although it is also trained to optimize primitive actions via the high-level advantage $A^h$), whereas $\pi_{k=1}^{d=1}$ corresponds to the low-level policy, primarily optimizing actions (but also trained to optimize next states $s_{t+1}$ as an immediate subgoal using $A^\ell$). This shared parameterization enables structured policy composition, where $\pi_{k=0}^{d=1} \circ \pi_{k=1}^{d=1}$ denotes a standard two-layer hierarchical rollout, and $\pi_{k=0}^{d=2}$ captures a shortcut inference from the top layer task goal to immediately executable action via shortcut policy distillation.

To encode the discrete hierarchy indicators $(\mathtt{k}, \mathtt{d})$, we reuse the same temporal encoding architecture developed for the flow-based policy in SSCQL (Appendix A.4). Specifically, each discrete input is passed through a learnable Fourier feature encoder that maps a scalar $x$ to $[\cos(2\pi x W), \sin(2\pi x W)]$, where $W \in \mathbb{R}^{\mathtt{dim}/2 \times 1}$ are learnable frequencies. These features are processed by separate two-layer MLPs with Mish activation and hidden size $\mathtt{dim}$, producing embeddings that are then concatenated with the state and goal embeddings to inform the policy network. This design enables the policy to condition flexibly on both temporal and structural context, supporting unified inference across hierarchical levels.

---

**Algorithm 3** Goal-Conditioned Single-Step Completion Policy (GC-SSCP)

---

1: **Initialize:** Policy parameters $\theta$, critic parameters $\phi_V$
2: Initialize target networks: $\theta' \leftarrow \theta$, $\phi'_V \leftarrow \phi_V$
3: **for** $N$ gradient steps **do**
4:      Sample batch $\{(s_t, a_t, r_t, s_{t+1}, g, g')\} \sim \mathcal{D}$

5:      **Critic Update:**                                                $\triangleright$ see Eq. 31
6:      Compute target: $\hat{q}(s_t, g) = r_t + \gamma V_{\phi'_V}(s_{t+1}, g)$
7:      Compute critic loss:

$$\mathcal{L}_V(\phi_V) = \mathbb{E}_{(s_t, s_{t+1}, g) \sim \mathcal{D}} \left[ L_\xi^2 \left( \hat{q}(s_t, g) - V_{\phi_V}(s_t, g) \right) \right]$$

8:
$$\phi_V \leftarrow \phi_V - \eta_\phi \nabla_{\phi_V} \mathcal{L}_V(\phi_V)$$

9:      **Actor Update:**
10:     *Advantage Estimates:*                                        $\triangleright$ see Eq. 37- 38

$$\hat{A}^h(s_t, g', g) = V_{\phi_V}(g', g) - V_{\phi_V}(s_t, g)$$

$$\hat{A}^\ell(s_t, a_t, g') = V_{\phi_V}(s_{t+1}, g') - V_{\phi_V}(s_t, g')$$

11:     *AWR Loss:*                                            $\triangleright$ see Eq. 34- 35

$$\mathcal{L}_0(\theta) = \mathbb{E}_\mathcal{D} \left[ -\exp\left( \beta \cdot \hat{A}^h(s_t, g', g) \right) \log \pi_\theta(a_t, g' \mid s_t, g, \mathtt{k}=0, \mathtt{d}=1) \right]$$

$$\mathcal{L}_1(\theta) = \mathbb{E}_\mathcal{D} \left[ -\exp\left( \beta \cdot \hat{A}^\ell(s_t, a_t, g') \right) \log \pi_\theta(a_t, s_{t+1} \mid s_t, g', \mathtt{k}=1, \mathtt{d}=1) \right]$$

12:     *Shortcut Completion Loss:*                          $\triangleright$ Distillation; see Eq. 36

$$\mathcal{L}_s(\theta) = \mathbb{E}_{(a_t, s_{t+1}) \sim \pi_{\mathtt{k}=0}^{\mathtt{d}=1} \circ \pi_{\mathtt{k}=1}^{\mathtt{d}=1}} \left[ -\log \pi_\theta(a_t, s_{t+1} \mid s_t, g, \mathtt{k}=0, \mathtt{d}=2) \right]$$

13:     *Total Actor Loss:*
$$\mathcal{L}_\pi(\theta) = \mathcal{L}_0(\theta) + \mathcal{L}_1(\theta) + \lambda \mathcal{L}_s(\theta)$$

14:
$$\theta \leftarrow \theta - \eta_\theta \nabla_\theta \mathcal{L}_\pi(\theta)$$

15:      **Target Network Updates:**
16:      $\phi'_V \leftarrow \mathrm{T}_\phi \cdot \phi_V + (1 - \mathrm{T}_\phi) \cdot \phi'_V$
17:      $\theta' \leftarrow \mathrm{T}_\theta \cdot \theta + (1 - \mathrm{T}_\theta) \cdot \theta'$
18: **end for**

---

### B.4.3 ALGORITHM AND HYPERPARAMETERS

The Goal-Conditioned Single-Step Completion Policy (GC-SSCP) framework is outlined in Algorithm 3. We implement our method using the JAX machine learning library, utilizing the OGbench benchmark suite[5] for standardized environment setups and baseline implementations. The common hyperparameters used across all goal-conditioned training environments are summarized in Table 13.

Across tasks/datasets, we vary the discount factor ($\gamma$) and the subgoal step interval for sampling the subgoals. The values are tabulated in Table 14. A discount factor of 0.99 performs well in medium-scale environments, while more complex or larger environments benefit from a slightly higher value of 0.995. Similarly, medium tasks use a subgoal step size of 25, whereas larger environments employ 50 or 100 steps. We present a detailed ablation study on subgoal step sensitivity in Section B.6.

---

[5] https://github.com/seohongpark/ogbench

Table 13: Common Hyperparameters for GC-SSCP Across All Offline GCRL Experiments

| Hyperparameter | Value |
|---|---|
| Batch Size | 1024 |
| Soft Update Rate for Q-Networks ($T_\phi$) | 0.005 |
| Soft Update Rate for Actors ($T_\theta$) | 0.001 |
| Learning Rates for All Parameters | $3 \times 10^{-4}$ |
| Hierarchy Level Dimension | 128 |
| Goal Representation Dimension | 10 |
| Training Steps | 500k |
| Actor Network Size | (512, 512, 512) |
| Critic Network Size | (512, 512, 512) |
| IQL Expectile ($\xi$) | 0.7 |
| AWR Inv. Temperature ($\beta$) | 5.0 |
| Shortcut Loss Coefficient ($\lambda$) | 1.0 |

Table 14: Hyperparameters for GC-SSCP across OGBench Environments

| Environment | Discount Factor ($\gamma$) | Subgoal Steps |
|---|---|---|
| pointmaze-medium-navigate-v0 | 0.99 | 25 |
| pointmaze-large-navigate-v0 | 0.995 | 100 |
| pointmaze-giant-navigate-v0 | 0.995 | 50 |
| pointmaze-teleport-navigate-v0 | 0.995 | 50 |
| pointmaze-medium-stitch-v0 | 0.99 | 25 |
| pointmaze-large-stitch-v0 | 0.995 | 50 |
| pointmaze-giant-stitch-v0 | 0.995 | 50 |
| pointmaze-teleport-stitch-v0 | 0.995 | 50 |
| antmaze-medium-navigate-v0 | 0.99 | 25 |
| antmaze-large-navigate-v0 | 0.995 | 100 |
| antmaze-giant-navigate-v0 | 0.995 | 100 |
| antmaze-teleport-navigate-v0 | 0.995 | 50 |
| antmaze-medium-stitch-v0 | 0.99 | 25 |
| antmaze-large-stitch-v0 | 0.995 | 50 |
| antmaze-giant-stitch-v0 | 0.995 | 50 |
| antmaze-teleport-stitch-v0 | 0.995 | 50 |

## B.5 TRAINING CURVES

Figure 9 presents training curves for offline goal-conditioned RL on OGBench environments across two task families: PointMaze and AntMaze, each with Navigate and Stitch variants. For each variant, the results are shown for four environments: *Medium*, *Large*, *Giant*, and *Teleport*. For each of these environments, the evaluation is done across five different goals. The vertical axis indicates the mean success rate averaged over 8 training seeds, with shaded regions representing the standard deviation. The horizontal axis shows training steps from 0 to 500k. These curves highlight the performance and learning stability of our method across progressively more complex multi-goal tasks.

## B.6 ABLATION

**AWR Inverse Temperatures.** Although we use the same inverse temperature hyperparameter ($\beta$) for both AWR losses defined in Equations 34 and 35, it is not strictly necessary. By default, we set $\beta = 5.0$ for both losses. To study the effect of this design choice, we perform a coarse hyperparameter sweep over $\beta_1$ (for the high-level controller) and $\beta_2$ (for the low-level controller), using values $\{0, 1, 5, 10\}$.

Figure 10 shows the results in the form of a heatmap, with $\beta_1$ along the vertical axis and $\beta_2$ along the horizontal axis. The values represent the average success rate, computed over 50 evaluation seeds and 5 different goals. We omit multiple training seeds here due to computational constraints, as this

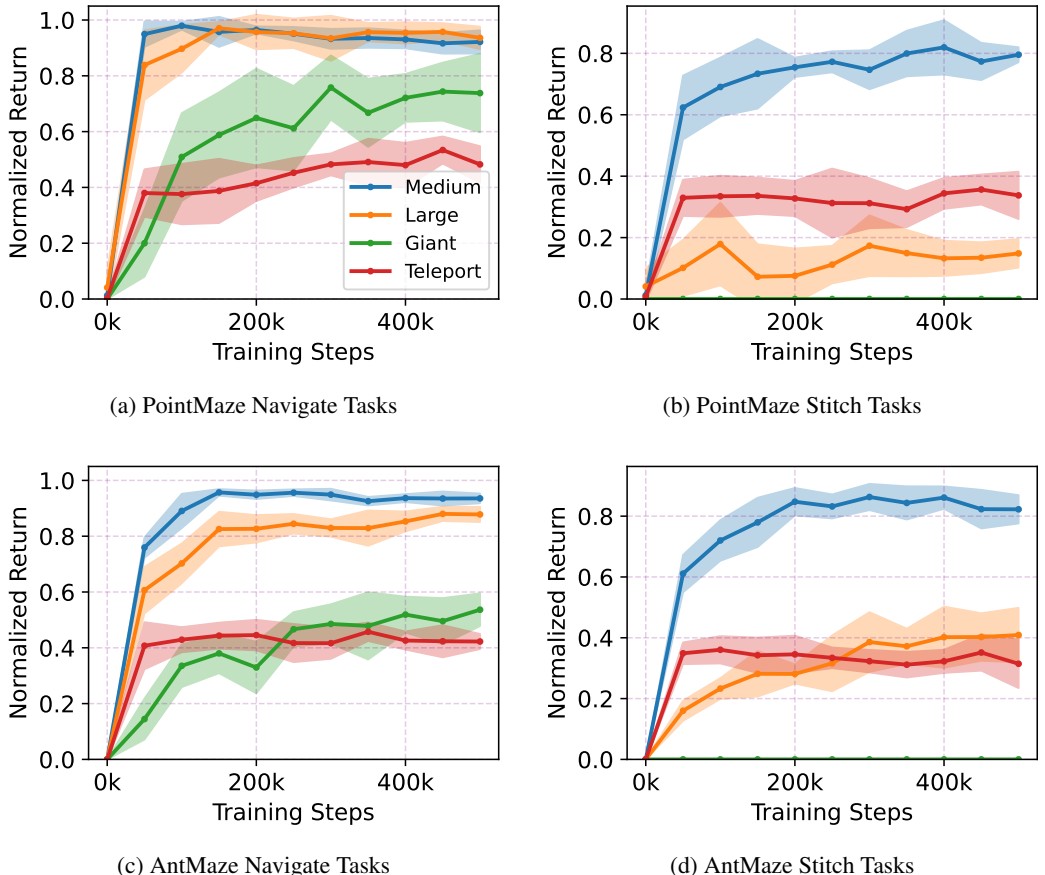

Figure 9: Training Curves for Offline Goal Conditioned RL in OGBench Environments with Multi-goal Tasks

would require several repeated runs and evaluations per setting. While our default hyperparameter setting $\beta_1 = \beta_2 = 5$ performs well, certain other combinations yield slightly better results as well.

Notably, when either $\beta_1$ or $\beta_2$ is set to 0, the corresponding AWR loss reduces to a standard behavior cloning loss. We observe that setting $\beta_1 = 0$ tends to degrade performance more significantly than setting $\beta_2 = 0$, suggesting that learning effective subgoals at the high level is more critical than fine-tuning low-level actions. Even when the low-level controller is trained with behavior cloning alone, the model can perform reasonably well if the high-level subgoal predictions are sufficiently informative.

**Inference Type.** Figure 11a presents an ablation on inference strategies. We compare the proposed flat inference $\pi_{k=0}^{d=2}$ against hierarchical inference $\pi_{k=0}^{d=1} \circ \pi_{k=1}^{d=1}$ on PointMaze Teleport Navigate and AntMaze Large Navigate. The training curves show no significant performance gap, suggesting that our hierarchy distillation effectively flattens the policy without sacrificing performance.

**Subgoal Choice.** Figure 11b studies the effect of the subgoal step interval, with values ranging from 5 to 500 on the x-axis and mean success rate on the y-axis. Both environments exhibit relatively stable performance across different subgoal intervals, with a slight peak around 25–100 steps. Even at the extremes (5 or 500), the performance does not degrade substantially, indicating robustness to this hyperparameter.

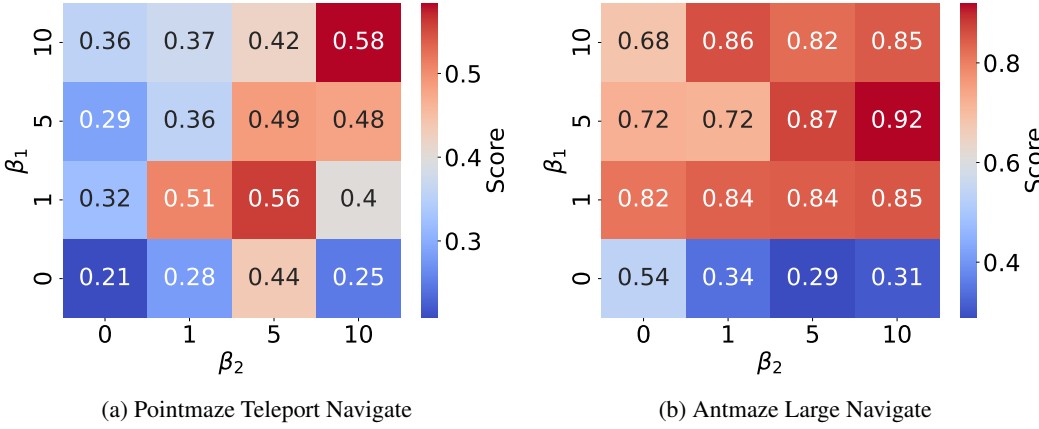

(a) Pointmaze Teleport Navigate

(b) Antmaze Large Navigate

Figure 10: Ablation on $\beta_1$ and $\beta_2$ hyperparameters for Offline GCRL tasks

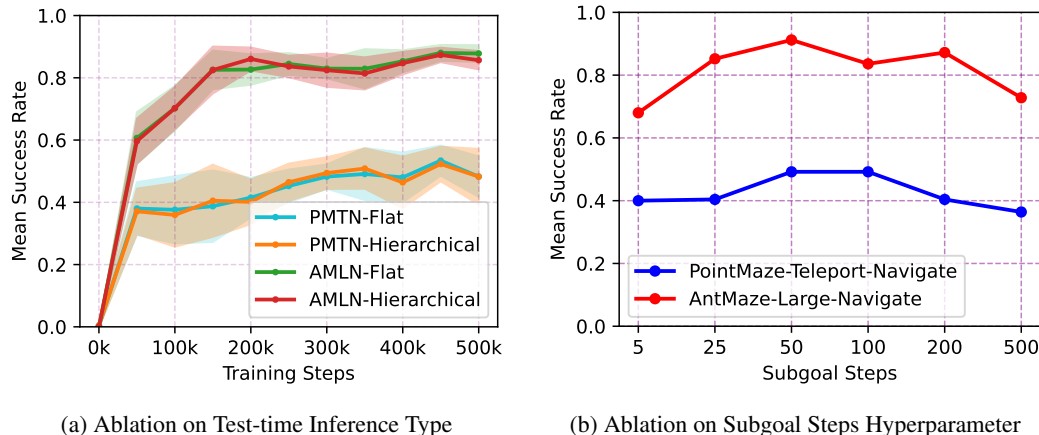

(a) Ablation on Test-time Inference Type

(b) Ablation on Subgoal Steps Hyperparameter

Figure 11: Ablation study based on inference type and subgoal steps for Offline GCRL tasks

## C  BEHAVIOR CLONING

The experiments in this section are based on the Behavior Cloning (BC) benchmark, specifically the *State Policy* setting introduced in Diffusion Policy Chi et al. (2024). We closely follow the experimental setup and implementation provided in the official Diffusion Policy GitHub repository[6]. Among the two baseline models provided—CNN-based and Transformer-based—we adopt the Transformer-based version due to its superior performance on the state-based robotic manipulation tasks considered. All implementations are in PyTorch. For a fair comparison, we adapt our Single-Step Completion Policy (SSCP) to match the temporal context and action sequence conditioning used by Diffusion Policy. SSCP is trained to predict future action sequences based on a history of past states, aligning with the context modeling and planning regime of Diffusion Policy.

We evaluate SSCP on behavior cloning tasks using state-based, proficient human demonstration datasets from RoboMimic (Mandlekar et al., 2021) and the contact-rich Push-T task (Florence et al., 2022). RoboMimic consists of five robotic manipulation tasks with high-quality human demonstrations, while Push-T requires precise manipulation of a T-shaped object under variable initial conditions and complex contact dynamics. In addition to Diffusion Policy, other baseline methods include LSTM-GMM (L-G) Mandlekar et al. (2021), Implicit Behavior Cloning (IBC) Florence et al. (2022), and Behavior Transformer (BET) Shafiullah et al. (2022). The results are reported in Table 15.

---

[6]https://github.com/real-stanford/diffusion_policy

While Diffusion Policy achieves strong results on a range of challenging manipulation tasks, it relies on iterative denoising with typically 100 reverse steps, resulting in significant inference-time latency. This poses a critical limitation in real-time robotics applications, where responsiveness is essential. In contrast, SSCP generates action predictions in a single forward pass, completely eliminating iterative sampling at inference. As demonstrated in our experimental results, SSCP achieves competitive performance while offering inference efficiency. This positions SSCP as a practical alternative for real-time deployment in robotics and controls.

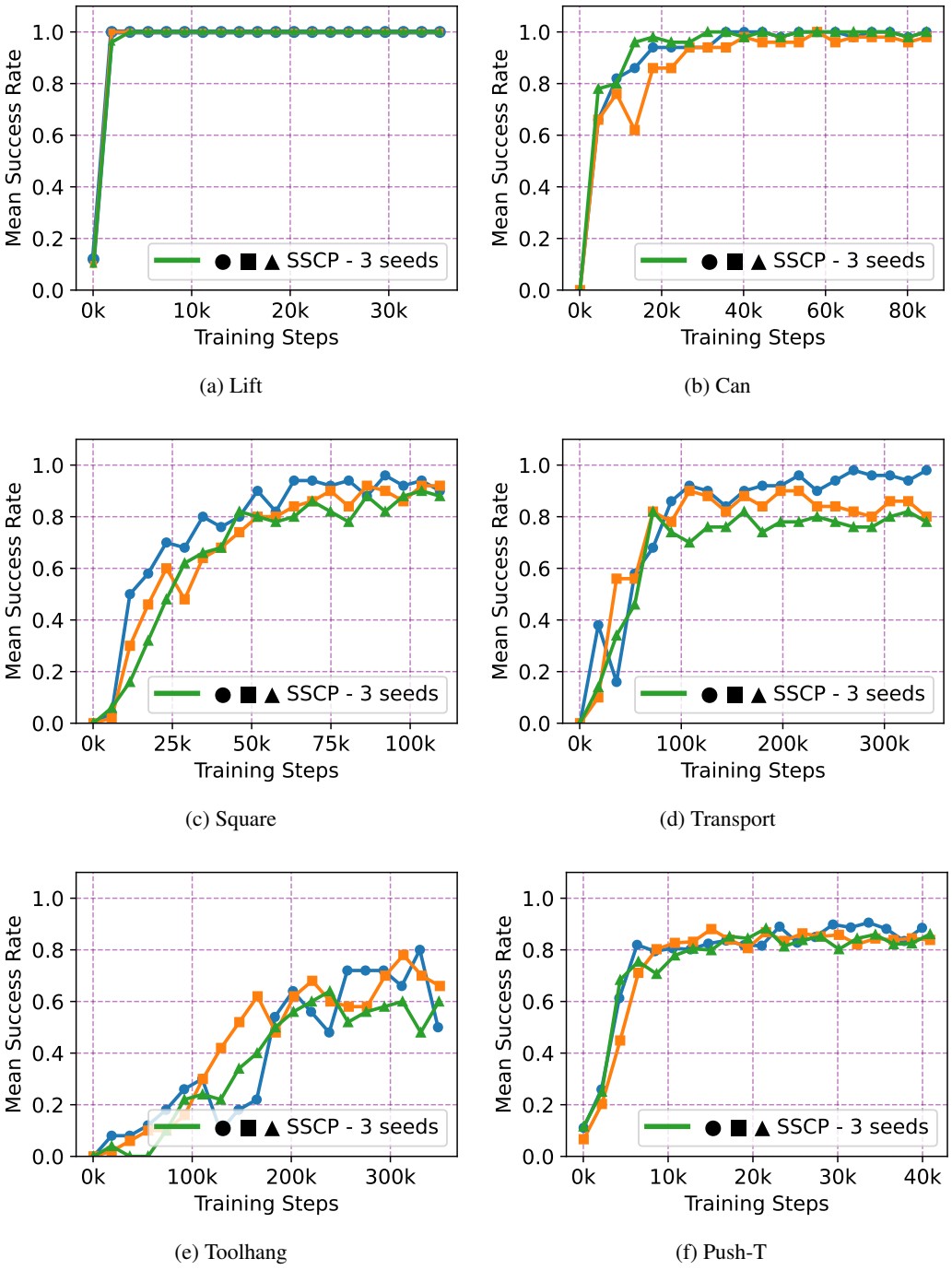

Figure 12: Training Curves for Behavior Learning Experiments (1000 epochs of training and evaluations with different seeds)

The training curves of SSCP on the imitation learning benchmark are presented in Figures 12a–f for the six robotic manipulation tasks: *Lift*, *Can*, *Square*, *Transport*, *Tool-Hang*, and *Push-T*. For each task, we train SSCP using three different random seeds for the model initialization and training, as well as distinct dataset seeds, ensuring diversity in data exposure and robustness to initialization. Each training run consists of 1000 epochs. To monitor learning progress and evaluate policy generalization, we periodically evaluate the model every 50 epochs during training. Each evaluation involves running 50 independent rollouts using environment seeds not seen during training. This setup enables us to report a reliable estimate of the model's performance and assess the stability of the learned policy across different training runs.

Following the evaluation protocol from Chi et al. (2024), we report the average performance over the last few checkpoints (in our case, the last 5) each averaged across 3 training seeds and evaluated over 50 different environment seeds (resulting in $3 \times 50 = 150$ rollouts per checkpoint, and a total of $150 \times 5 = 750$ rollouts per task). The results are summarized in Table 15. Note that while the baseline results taken from Chi et al. (2024) are the averages over the last 10 checkpoints, this choice is appropriate for their setting, as their models were trained for 4500 epochs. In our case, however, we only train for 1000 epochs, so averaging over the last 10 checkpoints would include models from the mid-training phase. Hence, using the last 5 checkpoints provides a more accurate and fair reflection of final policy performance in our setting.

Table 15: Behavior Cloning Final Results

| Tasks | DP | L-G | IBC | BET | SSCP |
|---|---|---|---|---|---|
| Lift | **1.00** | **0.96** | 0.41 | **0.96** | **1.00**±**0.00** |
| Can | **1.00** | 0.91 | 0.00 | 0.89 | **0.99**±**0.01** |
| Square | **0.89** | 0.73 | 0.00 | 0.52 | **0.90**±**0.03** |
| Transport | **0.84** | 0.47 | 0.00 | 0.14 | **0.86**±**0.08** |
| Toolhang | **0.87** | 0.31 | 0.00 | 0.20 | 0.64±0.09 |
| Push-T | 0.79 | 0.61 | **0.84** | 0.70 | **0.85**±**0.03** |
| Average | **0.88** | 0.68 | 0.21 | 0.57 | **0.87** |

In addition to reporting the average over the last checkpoints, Chi et al. (2024) also reports the maximum performance achieved by each method during the entire 4500 epochs of training, evaluated every 50 epochs. Although we train SSCP for only 1000 epochs, we follow a similar protocol and report its maximum performance in Table 16. Naturally, SSCP lags somewhat behind diffusion policy in this setting, given its one-shot inference (compared to diffusion policy's 100-step iterative denoising) and shorter training duration. Nonetheless, SSCP remains competitive with diffusion policy and significantly outperforms all other baselines included in the comparison.

Table 16: Behavior Cloning Results (Max Performance)

| Tasks | DP | L-G | IBC | BET | SSCP |
|---|---|---|---|---|---|
| Lift | **1.00** | **1.00** | 0.79 | **1.00** | **1.00** |
| Can | **1.00** | **1.00** | 0.00 | **1.00** | **1.00** |
| Square | **1.00** | **0.95** | 0.00 | 0.76 | 0.93 |
| Transport | **1.00** | 0.76 | 0.00 | 0.38 | 0.90 |
| Toolhang | **1.00** | 0.67 | 0.00 | 0.58 | 0.74 |
| Push-T | **0.95** | 0.67 | 0.90 | 0.79 | 0.89 |

The key hyperparameters used in our experiments are summarized in Table 17. The architecture and hyperparameters used in our behavior cloning experiments are largely consistent with those of the original Diffusion Policy (Chi et al., 2024). This similarity demonstrates that SSCP can be easily implemented on top of existing diffusion-based robot learning pipelines, achieving reliable performance with minimal architectural and hyperparametric modifications.

Table 17: Hyperparameters used in Behavior Cloning Experiments

| Hyperparameter | Value |
|---|---|
| Observation Horizon | 2 |
| Action Horizon | 8 |
| Action Prediction Horizon | 10 |
| Transformer Token Embedding Dimension | 256 |
| Transformer Attention Dropout Probability | 0.3 (0.01 for Push-T) |
| Learning Rate | 3e-4 (1e-3 for Toolhang) |
| Weight Decay | 1e-10 |
| Training Epochs | 1000 |
| Evaluation Frequency (in Epochs) | 50 |

# D BENCHMARK, TASKS AND BASELINES DETAILS

## D.1 OFFLINE RL

### D.1.1 D4RL DATASETS.

D4RL[7] (Datasets for Deep Data-Driven Reinforcement Learning) is a widely-used benchmark suite for offline reinforcement learning, where policies are learned solely from fixed datasets without further interaction with the environment. These datasets are paired with standardized Gym environments[8] (Fu et al., 2020; Brockman et al., 2016). We evaluate our method on continuous control tasks from the Gym-MuJoCo[9] locomotion suite (Todorov et al., 2012), including `HalfCheetah` (2D quadruped running), `Hopper` (2D monoped hopping), and `Walker2d` (2D biped walking). We consider three dataset types from D4RL: `medium`, `medium-expert`, and `medium-replay`. The `medium` datasets are generated by policies with intermediate performance trained using Soft Actor Critic (Haarnoja et al., 2018). The `medium-expert` datasets combine data from both medium and expert-level policies. The `medium-replay` datasets consist of replay buffers collected throughout the training of medium-level policies.

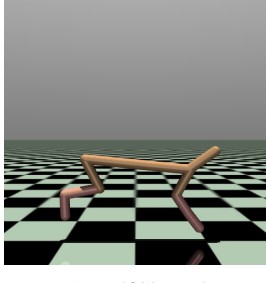 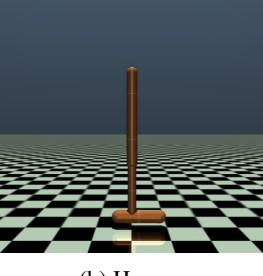 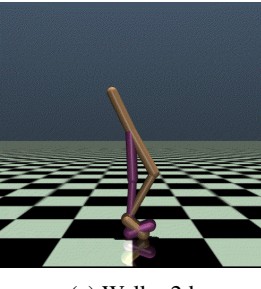

(a) HalfCheetah      (b) Hopper      (c) Walker2d

Figure 13: Gym Mujoco 2D Locomotion Continuous Control Environments

### D.1.2 BASELINES.

We compare SSCQL against a broad spectrum of baselines, encompassing both unimodal Gaussian policy methods and recent state-of-the-art generative approaches.

**Implicit Q-Learning (IQL)** (Kostrikov et al., 2021) addresses the offline RL setting by decoupling policy learning from out-of-distribution actions. IQL trains a state-value function $V_\phi$ using an asymmetric loss that implicitly avoids extrapolation errors in the critic. It enables robust policy extraction via advantage-weighted regression while maintaining off-policy stability. **TD3+BC** (Fujimoto & Gu, 2021) combines the TD3 actor-critic framework (Fujimoto et al., 2018) with a behavioral

---

[7]https://github.com/Farama-Foundation/D4RL
[8]https://gymlibrary.dev/, https://gymnasium.farama.org/
[9]https://github.com/google-deepmind/mujoco

cloning loss to regularize the policy towards the dataset. This simple hybrid effectively constrains policy updates to the support of the dataset, preventing the selection of out-of-distribution actions and overestimation by the Q-function. **Conservative Q-Learning (CQL)** (Kumar et al., 2020) introduces a conservative penalty during Q-function training to lower the estimated values of unseen (out-of-distribution) actions. This encourages learned policies to remain close to the dataset's action distribution, reducing overestimation and improving reliability in offline RL.

**Diffuser** (Janner et al., 2022) formulates trajectory generation as a denoising diffusion process over full state-action sequences. During inference, the reverse diffusion is guided by the gradient of predicted cumulative rewards, enabling model-based trajectory optimization toward high-return plans. **Implicit Diffusion Q-Learning (IDQL)** (Hansen-Estruch et al., 2023) uses a diffusion model to generate action samples, which are then filtered via rejection sampling using a Q-function trained with IQL. This approach retains the expressiveness of generative models while leveraging the Q-function for policy selection. **Diffusion Q-Learning (DQL)** (Wang et al., 2022) integrates diffusion-based generative models directly into policy optimization. It treats the policy as a stochastic denoising process and enables policy improvement by backpropagating Q-function gradients through the generative sampling chain. DQL exemplifies a broader trend of incorporating iterative diffusion into reinforcement learning algorithms. **Consistency Actor-Critic (CAC)** (Ding & Jin, 2023) leverages consistency models Song et al. (2023) to accelerate diffusion policy execution. It trains few-step denoising models that approximate the final output of the diffusion process, enabling faster inference. Although CAC improves training and inference efficiency relative to DQL, it still remains slower and less scalable than our method (SSCQL).

## D.2 BEHAVIOR CLONING

### D.2.1 SIMULATION ENVIRONMENTS AND DATASETS

We mainly evaluate our method on state-based tasks adopted by Chi et al. (2024) from the Robomimic benchmark (Mandlekar et al., 2021), a large-scale suite for imitation learning and offline reinforcement learning. We focus on all five manipulation tasks using the `ph` (proficient human) datasets, which contain high-quality teleoperated demonstrations. The tasks include: `Lift`, where the robot must grasp and lift a small cube; `Can`, which involves placing a coke can from a bin into a smaller target bin; `Square`, where the robot must insert a square nut onto a rod; `Transport`, a dual-arm task requiring the coordinated retrieval, handoff, and placement of a hammer while removing a piece of trash; and `Tool Hang`, where the robot assembles a hook-frame and hangs a wrench by performing precise, rotation-heavy manipulations. We additionally evaluate on the `Push-T` task, which involves pushing a T-shaped block to a fixed target using a circular end-effector. The task is contact-rich and demands fine-grained control.

### D.2.2 BASELINES

**Diffusion Policy** (Chi et al., 2024) adopts a behavior cloning framework by generating short-horizon action trajectories conditioned on past observations via a conditional denoising diffusion process. This enables receding-horizon planning and allows modeling of complex, temporally coherent, and multimodal behaviors, making it well-suited for high-dimensional imitation learning from raw demonstrations. The iterative denoising approach gracefully handles multimodal action distributions and offers stable training dynamics. **BC-RNN** (Mandlekar et al., 2021) is a behavior cloning variant that employs recurrent neural networks (RNNs) to capture temporal dependencies in sequential data. The policy is trained on sequences of states and actions, predicting actions autoregressively. Chi et al. (2024) reimplemented this approach using LSTMs combined with Gaussian mixture models (**LSTM-GMM**) to better represent multimodal action distributions. **Implicit Behavior Cloning (IBC)** (Florence et al., 2022) trains implicit policies using energy-based models (EBMs), which represent complex, potentially multimodal action distributions without requiring explicit likelihoods. This contrasts with standard explicit behavior cloning models trained via mean squared error on dataset actions, offering greater flexibility at the cost of more challenging optimization. **Behavior Transformer (BET)** (Shafiullah et al., 2022) was designed to model multimodal, unlabeled human behavior using transformer architectures. By autoregressively predicting action sequences with attention mechanisms, BET effectively captures long-range temporal dependencies.

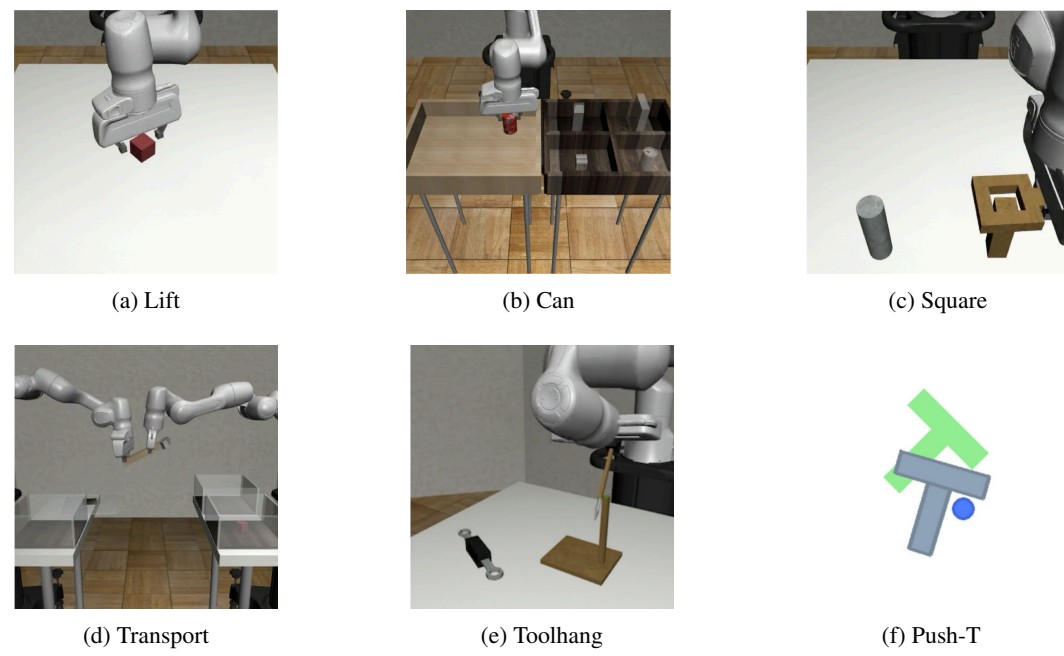

(a) Lift          (b) Can          (c) Square

(d) Transport          (e) Toolhang          (f) Push-T

Figure 14: Benchmark Tasks for Behavior Cloning Experiments

### D.3 OFFLINE GOAL CONDITIONED RL

#### D.3.1 OGBENCH

We evaluate on OGBench (Park et al., 2024a), a recent benchmark for offline goal-conditioned reinforcement learning (GCRL) that tests multi-task policy learning across diverse navigation and control scenarios. We consider the PointMaze and AntMaze environments based on the MuJoCo simulator (Todorov et al., 2012), where agents must learn both high-level navigation and low-level locomotion from diverse, static datasets. PointMaze involves controlling a 2D point mass, while AntMaze requires quadrupedal control with an 8-DoF Ant robot.

We consider 16 environment-dataset combinations: {pointmaze, antmaze} × {medium, large, giant, teleport} × {navigate, stitch}. Maze sizes vary from medium (smallest) to giant (twice the size of large). The teleport maze introduces stochasticity via black holes that transport the agent to randomly selected white holes, one of which is a dead end, penalizing the 'lucky' optimism and requiring robust planning.

Dataset types include navigate, collected via a noisy goal-conditioned expert exploring the maze, and stitch, composed of short goal-reaching trajectories that require policy stitching to reach distant goals. All datasets are generated with a low-level directional policy trained via SAC (Haarnoja et al., 2018). Following the OGBench evaluation protocol, we report average success rates over 6000 evaluation episodes (8 training seeds × 50 test seeds × 5 goals × 3 final evaluation epochs). Baseline results are taken directly from the OGBench benchmark paper, where each GCRL method is trained for 1M steps and evaluated every 100k steps, with final performance averaged over the last three evaluations (800k, 900k, and 1M steps). For GC-SSCP, we train for 500k steps and evaluate every 50k steps, reporting averages over the final three evaluations (400k, 450k, and 500k steps).

#### D.3.2 BASELINES

**Goal-Conditioned Behavioral Cloning (GCBC)** (Lynch et al., 2020; Ghosh et al., 2019) performs behavioral cloning using future states in the same trajectory as goals. It learns a goal-conditioned policy by treating the task as supervised learning over $(s, g, a)$ triplets, where $g$ is a future state in the trajectory. **Goal-Conditioned Implicit {V, Q}-Learning (GCIVL and GCIQL)** are goal-

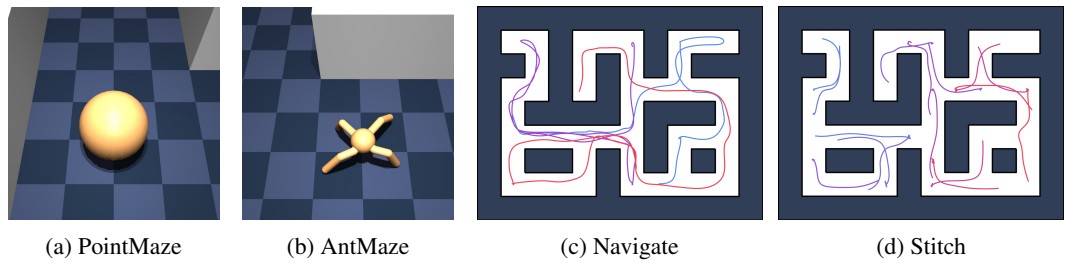

| (a) PointMaze | (b) AntMaze | (c) Navigate | (d) Stitch |

Figure 15: OGBench Agent and Dataset Types

conditioned extensions of Implicit Q-Learning (IQL) (Kostrikov et al., 2021), an offline RL algorithm that fits conservative value functions via expectile regression. GCIQL learns both $Q(s, a, g)$ and $V(s, g)$, while GCIVL is a simpler $V$-only variant that omits learning $Q$. **Quasimetric RL (QRL)** (Wang et al., 2023) is a value-learning algorithm that models the shortest-path distance between states as a *quasimetric*, an asymmetric distance function. It explicitly enforces the triangle inequality to better capture the geometric structure of goal-reaching tasks. **Contrastive RL (CRL)** (Eysenbach et al., 2022) is a goal-conditioned RL method that performs one-step policy improvement using a goal-conditioned value function trained via a contrastive learning objective. **Hierarchical Implicit Q-Learning (HIQL)** (Park et al., 2023) extracts a two-level hierarchical policy—consisting of a high-level subgoal predictor and a low-level action predictor—from a single goal-conditioned value function. At test time, inference is performed hierarchically: the high-level policy first predicts a subgoal, which the low-level policy then uses to generate executable actions.

