# OpenReview forum: "Flow-Based Single-Step Completion for Efficient and Expressive Policy Learning"
_ICLR.cc/2026/Conference — ICLR 2026 Poster_

### Official Review · Reviewer_SYWJ · 2025-10-31

**Soundness:** 3
**Presentation:** 3
**Contribution:** 3
**Rating:** 4
**Confidence:** 3

**Summary:**

This paper introduces Single-Step Completion Policy (SSCP), a flow-based generative policy framework for reinforcement learning that achieves single-step action generation while maintaining the expressiveness of multi-step generative models. The key innovation is training a completion model (instead of a shortcut model[1]) that predicts normalized completion vectors from intermediate flow points directly to target actions, bypassing iterative sampling. The authors demonstrate SSCP's effectiveness across three settings: (1) offline RL with behavior-constrained policy gradients (SSCQL), (2) goal-conditioned RL where hierarchical policies are distilled into flat inference (GC-SSCP), and (3) behavior cloning. The method achieves competitive or superior performance compared to diffusion-based baselines while offering substantial computational advantages.

[1] Frans, Kevin, et al. "One step diffusion via shortcut models." arXiv preprint arXiv:2410.12557 (2024).
[2] Park, Seohong, Qiyang Li, and Sergey Levine. "Flow q-learning." arXiv preprint arXiv:2502.02538 (2025).
[3] Espinosa-Dice, Nicolas, et al. "Scaling Offline RL via Efficient and Expressive Shortcut Models." arXiv preprint arXiv:2505.22866 (2025).
[4] Sheng, Juyi, et al. "MP1: MeanFlow Tames Policy Learning in 1-step for Robotic Manipulation." arXiv preprint arXiv:2507.10543 (2025).

**Strengths:**

1. Novel and well-motivated approach: The completion vector formulation elegantly addresses a fundamental limitation of diffusion/flow policies—the need for iterative sampling—while maintaining expressiveness for multimodal action distributions. Unlike bootstrap-based shortcut methods [1], SSCP uses ground-truth targets from the dataset, avoiding early training instability.
2. A significant practical advantage is that SSCP enables training generative policies without backpropagating through iterative generation chains, removing the requirement for distillation as shown in FQL
3. The paper demonstrates consistent improvements across diverse benchmarks:
4. The extension to goal-conditioned RL (GC-SSCP) is particularly innovative, showing that multi-level hierarchical reasoning can be compressed into a single flat policy without explicit hierarchical inference. This challenges the assumption that hierarchical structure is necessary for long-horizon tasks.
5. The paper provides extensive ablations on key hyperparameters (α₁, α₂), bootstrap targets vs. completion loss, and demonstrates the learned completion field can support both single-step and multi-step rollouts (Table 9).
6. Extensive comparisons with FQL in Appendix A.7

**Weaknesses:**

1. While the paper compares against FQL [Park et al., 2025], there are other recent few-step policy methods [3-4] that should be discussed and compared
2. The paper doesn't provide clear guidance on when SSCP is expected to outperform alternatives
3. While Table 9 shows multi-step rollout results, the analysis is limited
4. In Figure 7, the performance change could be quite large depending on hyperparameters chosen

**Questions:**

1. Why are multi-step actions worse in some cases, as shown in Table 9? Could you visualize the performance difference with x-axis = # of steps and y-axis = performance
2. How does SSCP compare with other one-step policies proposed recently, like MP1?
3. Can you explain why GC-SSCP fails as you scale up pointmaze-large-stitch to a larger setup?
4. It has been claimed by many papers extending FQL that distillation of the policy is the bottleneck, while no one has verified this approach. Can you verify this by simply training a flow policy and performing BPTT to see if we can achieve stronger performance by just directly optimizing the Q-value of the flow policy? If that is the case, it will strongly support the necessity of having a stronger policy than a distilled one.

---

> ### Author Response · Authors · 2025-11-17
>
> We sincerely appreciate the reviewer’s thorough and accurate summary of our contributions, as well as the detailed analysis of the paper’s strengths and limitations. We are grateful for the insightful recognition of the motivation behind the completion-policy formulation, its practical advantages over iterative generative policies, and the breadth of our empirical evaluations. We are also glad that the reviewer found the goal-conditioned extension innovative and liked the depth of the ablation studies conducted in our experiments. Below, we address the reviewer’s questions and concerns point by point.
>
> ### **Few-step policy methods:**
> We sincerely thank the reviewer for pointing out these recent few-step policy methods. Many of these works appeared during or after the preparation of our manuscript, and were therefore not available for direct comparison at the time. We will review the most relevant methods and incorporate a discussion of their relation to our approach in the final version to ensure proper contextualization. Some of these works are also discussed in response to other comments below.
>
> ### **SSCP vs. alternatives:**
> Thank you for raising this point. We address this question by discussing the pros, limitations, and trade-offs of our method in comparison to established alternatives. In our experience, SSCP exhibits stable performance as long as the primary hyperparameters ($\alpha_1$ and $\alpha_2$) are appropriately tuned. The method also benefits from large batch sizes, which accelerate training both in terms of gradient steps and wall-clock time, achieving up to a 64× speedup compared to DQL. Empirically, SSCP matches the performance of its undistilled multi-step counterpart (DQL), while GC-SSCP attains comparable results to hierarchical baselines such as HIQL, as reported in the results section.
>
> At the same time, we acknowledge that single-step completion models may have limitations in capturing highly multimodal or high-dimensional behaviors relative to multi-step generative policies. Similarly, in long-horizon goal-conditioned tasks, bypassing explicit hierarchical structures may occasionally constrain performance. These limitations are explicitly noted in the conclusion section. Overall, these points might be helpful in considering when to choose SSCP over the existing alternatives.
>
> ### **Table 9 multi-step rollout results:**
>
> We appreciate the reviewer’s feedback. Table 9 in the appendix is intended as a diagnostic reference to illustrate that, unlike FQL, SSCQL naturally supports multi-step flow rollouts during evaluation. This also provides a means to assess the quality of single-step distillation: if distillation is ideal, multi-step rollout performance should closely match single-step predictions. We will elaborate on this motivation and its implications in the final version. We also thank the reviewer for carefully examining these extended results.
>
> Overall, the results indicate that the learned completion vector captures a coherent flow structure, effectively guiding iterative action generation toward high-reward regions, even in the absence of explicit multi-step backpropagation during training.. This highlights that SSCQL can support efficient single-step inference by default, while also allowing flow-based rollouts when additional computation is feasible. Further discussion on scaling the number of flow rollout steps is provided below in response to a related question.
>
> ***(1/3)***

---

> > ### Author Response · Authors · 2025-11-17
> >
> > ### **Performance change with hyperparameters:**
> > We agree that the variations appear pronounced due to the wide (and sometimes unrealistic) hyperparameter ranges explored in the heatmap. For instance, we also include coefficient values of 0 that effectively remove BC regularization, a setting known to degrade offline RL performance due to distribution shift. As described in Appendix A.4, we fix $\alpha_1 = 1.0$ and tune only $\alpha_2$, which substantially simplifies tuning and corresponds to the results in Table 1.
> >
> > Nevertheless, we acknowledge that the two BC hyperparameters introduce additional tuning overhead compared to simpler baselines (e.g., TD3+BC) and explicitly note this as a limitation in Section 6. However, we believe this trade-off is reasonable given the improved stability and performance achieved by the proposed framework.
> >
> >
> >
> >
> > ### **Multi-step action generation:**
> > The observed performance differences primarily stem from the fact that the $\alpha$ hyperparameters were tuned specifically for single-step action generation, and the same learned completion model was used “as-is” for multi-step rollout without any additional re-tuning. As also noted in methods like FQL, tuning the BC regularization coefficient has a critical impact on rollout and overall performance.
> > The purpose of Table 9 is to illustrate that multi-step rollouts are feasible and to assess whether single-step and multi-step rollouts yield comparable results. While improvements are observed in certain tasks using the same completion vector fields (without further training or additional hyperparameter tuning), we do not claim that multi-step rollouts universally enhance performance within our framework.
> >
> > | **Number of Steps** | **HC-Medium** | **HC-Medium-Replay** |
> > | ------------------- | ------------- | -------------------- |
> > | 1                   | 52.7          | 44.8                 |
> > | 5                   | 52.6          | 46.9                 |
> > | 10                  | 53.4          | 47.1                 |
> > | 100                 | 54.5          | 47.7                 |
> >
> >
> >
> > ### **Comparison with other one-step policies:**
> >
> > We thank the reviewer for highlighting the recent one-step policy methods. MP1 appears to be a recent approach, but it does not report results on D4RL benchmarks, and its datasets are not publicly available, likely because it is an unpublished work in progress. We are happy to incorporate additional baselines when equitable comparisons are feasible and will explore ways to include such results before the final revision. Notably, some core contributions of MP1 (such as dispersive loss and architectures supporting conditioning on 3D point clouds) are orthogonal to our method and could potentially complement our completion modeling and Q-learning framework, as well as the flat inference goal-conditioned framework we propose.
> >
> > To further address the reviewer’s question, we evaluated another recent one-step policy method, SORL. While SORL involves multiple hyperparameters and the implementation details are less straightforward than FQL or SSCP, we conducted a sweep over its primary q_loss_coef hyperparameter with values {1, 10, 100}, based on recommendations from its reported tasks, and report the results below. We acknowledge that this hyperparameter sweep may not be fully exhaustive, but it provides a reasonable comparison for the purposes of this discussion.
> >
> >
> >
> > | - | **walker2d-medium** | **hopper-medium** | **halfcheetah-medium** |
> > | --------------- | ------------------- | ----------------- | ---------------------- |
> > |SORL(q_loss_coef= **100**) | 15.3                | 50.6              | 57.0                   |
> > |SORL(q_loss_coef= **10**)         | 82.2                | 47.8              | 46.7                   |
> > | SORL(q_loss_coef= **1**) | 70.2                | 43.9              | 43.8                   |
> > | SORL (best) | **82.2**               | 50.6             | **57.0**                   |
> > | SSCQL | **84.2**                | **102.4**             | 52.3                   |
> >
> >
> >
> > ### **Larger setups in pointmaze stitch tasks:**
> >
> > This is an important observation. Larger PointMaze-Stitch tasks are challenging due to their extended horizons, sparse intermediate rewards, and the need to stitch shorter trajectories to achieve long-horizon goal completion. These difficulties are well-documented in recent GCRL literature, and remain an open area of research. Consequently, even hierarchical inference methods, including those used as baselines, face similar challenges on these tasks.
> >
> > ***(2/3)***

---

> > > ### Author Response · Authors · 2025-11-17
> > >
> > > ### **BPTT and policy distillation:**
> > >
> > > We thank the reviewer for raising this important point. Conceptually, directly training a flow policy with BPTT for Q-value optimization resembles methods such as DQL, where a diffusion (a general form of flow matching) actor is optimized through deterministic off-policy policy gradients. Empirically, DQL generally outperforms FQL in D4RL tasks, consistent with the reviewer’s observation. This naturally raises the question: how does our distillation method differ from that in FQL?
> > >
> > > The distinction lies in how BPTT is handled and how policy gradients are applied in these two methods. In FQL, only the distilled MLP actor is optimized using policy gradients, while the flow-matching model itself is not directly optimized for Q-values, which inherently seems to limit its expressivity after distillation.
> > >
> > > In contrast, SSCQL applies deterministic policy gradients directly to the flow model while avoiding BPTT by design, since each completion vector produces actions in a single generative step from any intermediate step. In Figure 1 (main text of the manuscript), this corresponds to propagating gradients along the red “completion” lines rather than through the full sequence of violet “velocity” vectors in standard flow rollouts (similar in spirit to DQL). This design enables stable policy optimization without long gradient chains. We appreciate the reviewer highlighting this conceptual distinction and will include a detailed clarification in Appendix A.7.
> > >
> > >
> > > ***(3/3)***

---

### Official Review · Reviewer_o22J · 2025-10-31

**Soundness:** 4
**Presentation:** 4
**Contribution:** 3
**Rating:** 8
**Confidence:** 1

**Summary:**

The authors propose single-step completion in flow matching, enabling transitions between arbitrary intermediate states rather than only predicting instantaneous velocity (the zero-jump case). While the standard self-consistent shortcut model relies on bootstrapped, potentially inaccurate targets—risking drift and unstable exploration—the authors address this by fixing the target to the final sample and learning one-step completions from any time step to the final one

This leads to the Single-Step Policy Completion (SSPC) objective, which learns complex, multimodal policies in a single step. The method achieves competitive performance with significantly faster training and inference. They further extend it to goal-conditioned RL, where a shortcut-based flat policy replaces hierarchical structures, improving efficiency while maintaining strong results.

**Strengths:**

- Big efficiency gains while maintaining or exceeding baseline performance.
- The paper is well-written and the method is well documented.
- Figure 8 in the appendix shows the strength of SSCP against shortcut models and makes the case that relying on bootstrap targets induces instability in the training. **I think that this figure is important since it clearly motivates the use of SSCP and thus would like to see it in the main text.**

**Weaknesses:**

N/A

**Questions:**

N/A

---

> ### Author Response · Authors · 2025-11-17
>
> Thank you for the positive and thoughtful assessment of our work. We appreciate the clear summary of the core ideas behind SSCP and its advantages over other diffusion-based approaches. We are also glad that the reviewer highlighted the competitive performance of our method and its further extension to flatten the hierarchical structures in goal-conditioned RL settings.
>
> Regarding the suggestion to include Figure 8 in the main text, we agree that moving this result forward would strengthen the narrative and better motivate SSPC. We will carefully consider this recommendation when preparing the camera-ready version. While no concerns were raised by the reviewer, we are, of course, happy to address any additional questions or comments. We are encouraged to see the core motivations and contributions reflected precisely in the review.

---

> > ### Comment · Reviewer_o22J · 2025-11-22
> >
> > I acknowledge the authors response and would like to signal that I have no further questions or comments. Thank you!

---

### Official Review · Reviewer_a6ST · 2025-10-31

**Soundness:** 3
**Presentation:** 2
**Contribution:** 2
**Rating:** 6
**Confidence:** 3

**Summary:**

This work considers generative modeling in RL, like diffusion models and flow-matching. Although these models have made great progress, they always rely on high inference costs and training instability caused by iterative sampling. To address this, this work proposes the Single-Step Completion Policy (SSCP), with an augmented flow-matching objective to predict direct completion vectors from intermediate flow samples, enabling accurate, one-shot action generation. This method can be extended into offline RL, offline-to-online RL, and online RL. This method is verified across various settings and different environments.

**Strengths:**

- This paper is well written and easy to follow.

- Stable training and efficient sampling are core concerns in diffusion policies in RL.

- The proposed method is flexible to different settings, like offline RL, online RL, and offline-to-online RL.

**Weaknesses:**

- The Q update function (4) utilizes the standard TD error. However, various works in offline RL propose that there is an overestimation error of the Q function caused by the distribution shift. Thus, several works will choose conservative Q learning techniques like IQL. What about the performance of using IQL in the offline setting?

- What is the difference between online RL and offline RL when applying SSCP?

- In offline-to-online experiments like Fig.4 and Fig.5, it seems that online fine-tuning in various environments can not improve the performance. Is there any explanation?

- It is better to add offline-to-online and online RL experiments to the main text.

- There are still some diffusion policies for RL that need to be discussed, including online fine-tuning [1-3] and offline diffusion planners [4-6].

Ref:

[1] Policy agnostic RL: Offline RL and online RL fine-tuning of any class and backbone

[2] Exploratory Diffusion Model for Unsupervised Reinforcement Learning

[3] Efficient Online Reinforcement Learning for Diffusion Policy

[4] What makes a good diffusion planner for decision making?

[5] Simple hierarchical planning with diffusion

[6] Latent diffusion planning for imitation learning

**Questions:**

See weaknesses above.

---

> ### Author Response · Authors · 2025-11-17
>
> Thank you for the constructive review of our work. We are glad that you found the method clearly presented, the evaluations thorough, and the approach both performant and efficient. We appreciate your recognition of the simplicity our framework brings to offline RL. Below, we address your comments in detail.
>
> ### **Q function update in SSCQL:**
> Indeed, distribution shift in offline RL can lead to overestimation errors in Q-values, and conservative methods such as CQL and IQL mitigate this through conservative value estimation. However, another effective way to address distribution shift is to constrain the policy to generate actions close to those supported by the dataset, ensuring that Q-functions are queried only on in-distribution state–action pairs. Our method SSCQL is based on this approach of restricting the policy updates (equations 2, 17).
>
> We appreciate the reviewer noting that the proposed policy representation can also be coupled with other value-learning methods, such as IQL; however, this might only add further conservatism, as we already restrict the policy updates. For a comprehensive benchmarking, please note that we do include conservative value learning baselines like IQL and CQL in evaluation table 1. Furthermore, we include IQL and IDQL, two specific baselines that use IQL-based critics, but have differences in terms of using unimodal gaussian policy and diffusion-based policy, respectively.
>
> ### **SSCP - Online vs Offline RL:**
> Thank you for this insightful question, as it touches on the core of our framework. In offline RL, using dataset actions for completion targets inherently provides a behavior regularization effect, similar to that in behavior-constrained offline RL methods. The flow and completion losses effectively act as BC-style regularizers that keep generated actions near the dataset’s support (equivalence between equations 2 and 17), implicitly addressing the distribution shift and extrapolation issues noted in the previous question.
>
> In contrast, in online RL, the replay buffer is initially empty and is populated solely through ongoing data collection. Thus, behavior regularization (as in BC) is counterproductive to exploration. Instead, we encourage exploration through epsilon greedy strategy with gradually decreasing exploration. Furthermore, while large batch sizes can stabilize offline training, they can be less effective in online RL since agents must actively collect new experiences. For online adaptation, we use smaller batch sizes and learning rates with more gradient steps, details and results are provided in Appendix A.4 and Figure 6.
>
> In summary, the way SSCP is designed, it is currently better suited for offline RL; however, we demonstrate that it can be adapted to online learning with some modifications. We also highlight further exploration of this transfer as a future research direction in the conclusion section.
>
>
> ### **Offline-to-online experiments:**
> We agree that when the offline policy already performs well, there is naturally less room for improvement. Nonetheless, we note two important observations here:
> (1) Our method consistently preserves the performance achieved during offline training, demonstrating robustness under online finetuning; and
> (2) While the absolute gains may often appear modest, the relative improvements are meaningful (e.g., in the final column of table 4 - HC-M: ~14%, H-M: ~61%, HC-MR: ~30%, W2d-MR: ~24%).
>
>
> ### **Additional experiments in main text:**
> We appreciate this constructive suggestion. Due to space constraints, we deferred several experiments, including the offline-to-online and online RL results, to the appendix. We agree that including them in the main paper would make the presentation more complete. We will carefully consider the reviewer’s recommendation when preparing the camera-ready version, making use of the additional one page while also taking into account requested additions from other reviewers. We also thank the reviewer for their effort in examining the extended results.
>
>
>
> ### **Additional references:**
> Thanks for pointing out these recent works. We will review the suggested references and incorporate the most pertinent ones into the manuscript. While our focus and formulation somewhat differ from these methods, we acknowledge the conceptual connections: for example, SSCQL can be relevant to online finetuning, and GC-SSCP is related to goal-conditioned / hierarchical planning methods. These directions, although orthogonal to our primary contribution, can help position our work in the broader context of diffusion-based reinforcement learning research.

---

> > ### Comment · Reviewer_a6ST · 2025-11-25
> >
> > I have read the authors' response and other reviewers' comments. I keep my score positive for this work.

---

### Author Response · Authors · 2025-12-02

We thank all reviewers for their thoughtful feedback and encouraging comments. In this common response, we briefly restate our key contributions and summarize the changes incorporated in the revised version.

In this work, we introduce the Single-Step Completion Policy (SSCP), a generative policy trained with an augmented flow-matching objective to predict completion vectors directly from intermediate flow samples (Section 3). This enables accurate, one-shot action generation and removes the need for backpropagation through long computation chains (BPTT). In the offline RL setting, the completion-policy class is compatible with standard off-policy actor–critic methods (e.g., DDPG+BC) while avoiding multi-step flow rollouts during training and inference. As a result, SSCQL achieves performance competitive with diffusion-based baselines (DQL, CAC, IDQL) while being substantially faster in both training and evaluation, with similar gains observed in online, offline-to-online, and behavior-cloning settings.

We then extend the notion of completion/shortcut modeling to goal-conditioned RL to effectively exploit subgoal structures while enabling flat (non-hierarchical) inference (Section 4). The resulting method (GC-SSCP) compresses/flattens multi-step decision-making into a single-step inference, which makes it beneficial in multi-goal offline RL tasks where hierarchical methods have demonstrated performance. Our method not only significantly outperforms the flat baselines, but also surpasses hierarchical SOTA on average in OGBench tasks

**Changes in the revision:**

The initial submission placed the primary offline RL and offline GCRL results in the main text, with offline-to-online finetuning, online RL, BC benchmarks, and extensive ablations in the appendix due to page limits. Several reviewers encouraged elevating more of these results into the main paper. With one additional page permitted in the final version, we have moved key results (Tables 3, 4, and 5) into the main text, with newly added material highlighted in blue. Due to space constraints, some ablations remain in the appendix; however, we checked/strengthened cross-references from the main text to these sections and will further refine this in the camera-ready version.


We sincerely hope that our responses have addressed the reviewers’ concerns during the discussion phase. We are grateful for the constructive engagement from the reviewers and the AC, and hope that these revisions and clarifications will assist in the final evaluation.

---

### Meta-Review · Area_Chair_e3Wt · 2026-01-02

**Summary:**

This paper proposes a generative policy that learns a single-step completion from intermediate flow/diffusion states directly to the final action, aiming to retain the multimodal expressiveness of diffusion/flow policies while avoiding iterative sampling and long-horizon backpropagation. Reviewers largely agreed that the approach is well-motivated and delivers practical efficiency gains in both training and inference, with strong empirical performance demonstrated across offline RL, offline-to-online fine-tuning, online RL, behavior cloning, and an extension to goal-conditioned RL.

The rebuttal and revision effectively clarified several technical concerns and improved the paper’s presentation by moving key experimental results into the main text. Some related work on very recent one-/few-step diffusion methods for RL appeared concurrently with or after the submission deadline and therefore was not considered by the AC as a required point of comparison for this decision.

**Reviewer Concerns:**

The authors addressed the concerns about offline RL distribution shift and Q-function overestimation, the distinction between offline, online, and offline-to-online settings, and comparisons with very recent one-/few-step diffusion methods.

A remaining limitation is that it is not yet fully clear what expressive power or behavioral flexibility may be lost by removing iterative sampling, which warrants further investigation.

**Reviewer Scores:**

The first two reviewers are likely to maintain their initial scores, while the third reviewer would likely slightly increase their score after the discussion.

---

### Decision · Program_Chairs · 2026-01-26

Accept (Poster)